



# Extracting recent short-term glacier velocity evolution over Southern Alaska from a large collection of Landsat data

Bas Altena[1], Ted Scambos[2], Mark Fahnestock[3], and Andreas Kääb[1]

[1]Department of Geosciences, University of Oslo, Blindern, 0316 Oslo, Norway
[2]National Snow and Ice Data Center (NSIDC), University of Colorado, Boulder, CO 80303, USA
[3]Geophysical Institute, University of Alaska Fairbanks, Fairbanks, AK 99775, USA

*Correspondence to:* Bas Altena (bas.altena@geo.uio.no)

**Abstract.** The measurement of glacier velocity fields using repeat satellite imagery has become a standard method of cryospheric research. However, the reliable discovery of important glacier velocity variations on a large scale from noisy time-series of such data is still problematic. In this study we propose a new post-processing procedure for assembling a set of velocity fields in time-series that generates a better visualization of glacier speed changes when the velocity fields are sparse or noisy. We demonstrate this automatic method on a large glacier area in Alaska/Canada. The visualization tool provides an overview of where and when interesting glacier dynamics are occurring. The goal is not to improve accuracy or precision, but the timing and location of ice flow events such as glacier surges. Building upon existing glacier velocity products from the GoLIVE data set (https://nsidc.org/data/golive), we compile a multi-temporal stack of velocity data over the Saint Elias Mountain range and vicinity. Each layer has a time separation of 32 days, making it possible to observe details such as within-season velocity change over an area of roughly 600 000 km$^2$. Our methodology is robust as it is based upon a fuzzy voting scheme to filter multiple outliers. The multi-temporal data stack is then smoothed to facilitate interpretation. This results in a spatio-temporal dataset where one can identify short-term glacier dynamics on a regional scale. Our implementation is fully automatic and the approach is independent of geographical area or satellite system used. Within the Saint Elias and Kluane mountain ranges, several surges and their propagation characteristics are identified and tracked through time, as well as more complicated dynamics in the Wrangell's mountains.

## 1 Introduction

Alaskan glaciers have a high mass turn-over rate (Arendt, 2011) and they can contribute considerably to sea level rise (Gardner et al., 2013; Arendt et al., 2013). Many of the glaciers have been identified as surge-type, from direct observations or from their looped moraines (Post, 1969; Herreid and Truffer, 2016). Furthermore, topographic glacier change in this region is heterogeneous (Muskett et al., 2003; Berthier et al., 2010; Melkonian et al., 2014) which is another indication of complicated responses.




Gaining better understanding of the drivers that cause glacier mass re-distribution is therefore of great importance.

Glacier velocity monitoring through satellite remote sensing has proven to be a useful tool to observe velocity change on a basin scale. Several studies have focused on dynamics of individual glaciers in Alaska, at an annual or seasonal resolu-
tion (Fatland and Lingle, 2002; Burgess et al., 2012; Turrin et al., 2013; Abe and Furuya, 2015; Abe et al., 2016). Such studies can give a better understanding of the specific characteristics of a glacier, and which circumstances are of importance for this behaviour and response. Region-wide annual or "snapshot" velocities also have been estimated over the Saint Elias Mountain range in previous studies (Burgess et al., 2013; Waechter et al., 2015). Their results give a first-order estimate of the dynamics at hand. With frequent satellite data coverage, it is possible to detect the time of glacier speed-ups to within a week (Altena
and Kääb, 2017b), although the study did not include an automated approach. In the most recent work, regional analyses have been conducted with over sub-seasonal (Moon et al., 2014; Armstrong et al., 2017) and multi-decadal (Heid and Kääb, 2012; Dehecq et al., 2015) periods. With such data one is able to observe the behaviour of groups of glaciers that experience similar climatic settings. Consequently, surges and other glacier-dynamical events can be put into a wider spatio-temporal perspective. In this contribution we want to develop the methodological possibilities further and try to extract glacier velocities at a monthly
resolution over a large region. The presented method retains spatial detail present in the data and does not simplify the flow structure. Consequently, we want to improve knowledge about the influence and timing of tributary and neighbouring ice flow variations.

Since the launch of Landsat 8 in 2013 a wealth of high-quality medium-resolution imagery is being acquired over the
cryosphere on a global scale. Onboard data storage and rapid ground-system processing have made it possible to almost continuously acquire imagery. The archived data has enormous potential to advance our knowledge in glacier flow. Extraction of glacier velocity is one of the stated mission objectives (Roy et al., 2014). However, the data rate far exceeds the possibilities for manual interpretation. Fortunately, automatically generated velocity products are now available (Scambos et al., 2016; Rosenau et al., 2016), though at this point sophisticated quality control and post processing methods are still being developed.
Up to now, most studies of glacial velocity have had an emphasis on either spatial or temporal detail. When temporal detail is present, studies focus on a single or a handful of glaciers (Scherler et al., 2008; Quincey et al., 2011; Paul et al., 2017). On the other hand, when regional assessments are the focus, the temporal resolution ranges from a single time stamp up to annual resolution (Copland et al., 2009; Dehecq et al., 2015; Rosenau et al., 2015). Furthermore, most studies rely on filtering in the
post-processing of vector data by using the correlation (Scambos et al., 1992; Kääb and Vollmer, 2000) or through median filtering within a zonal neighborhood (Skvarca, 1994; Paul et al., 2015). Some sophisticated post-processing procedures are available (Maksymiuk et al., 2016), but rely on the coupling with models based upon the Navier-Stokes equations. Also geometric properties can be taken into account during the matching to improve robustness and reduce post-processing efforts, such as reverse-correlation (Scambos et al., 1992; Jeong et al., 2017) or triangle closure (Altena and Kääb, 2017a).



Thus glacier velocity data is increasingly available, but in general post-processing is not at a sufficient level to directly exploit the full information content within these products. In this study we aim at the construction of a post-processing chain that is capable of extracting temporal information from stacks of noisy velocity data. Our emphasis is on discovering patterns over a mountain-range scale. Analysis of the individual details of the glacier-dynamical patterns identified by the processing

will be considered in later work. For a single glacier, a manual selection of low-noise, good-coverage velocity data sets is possible. However, such a strategy will not be efficient when multiple glaciers or mountain ranges are of interest. Therefor, our implementation focuses on automatic post-processing, without the help of expert knowledge or human interaction.

In this study, we discuss the data used and provide background on the area under study. We then introduce the spatio-

temporal structure of the data, followed by an explaination of our process for vector "voting" and vector field smoothing. The next section highlights our results and our validation and assessment of the performance of our method.

## 2   Data and study region

### 2.1   GoLIVE velocity fields

The velocity fields used in this study are based upon repeat optical remote sensing imagery and are distributed through the National Snow and Ice Data Center (NSIDC) (Scambos et al., 2016). These velocity fields are derived from finding displacements between pairs of Landsat 8 imagery, using the panchromatic band with 15 meter resolution. A high-pass filter of one kilometer spatial scale is applied before processing. Normalized cross-correlation is applied between the image pairs on a sampling grid with 300 meters spacing (Fahnestock et al., 2016; Scambos et al., 1992) and a template size of 20 pixels (or 300 m). The

resulting products are grids with lateral displacements, the absolute correlation value, signal-to-noise ratio and ratio between the two best matches. For a detailed description of the processing chain see Fahnestock et al. (2016).

The Landsat 8 satellite has a same-orbit revisit time of 16 days and a swath width of 185 km. Only scenes which are at least 50% cloud-free are used (as determined by the provided estimate in the metadata for the scenes). Consequently, not every

theoretical pair combination is matched, and also pairs across tracks are neglected (cf. (Altena and Kääb, 2017a)) to avoid more complicated viewing geometry adjustments. Georeference errors are compensated by the estimation of a polynomial bias surface through areas outside glaciers (i.e., assumed stable). The glacier mask used for that purpose is from the Randolp Glacier Inventory (RGI) (Pfeffer et al., 2014). The resulting grids come in Universal Transverse Mercator (UTM) projection and if orthorectification errors are minimal, displacements for precise georeferencing require only horizontal movement of a

few meters (generally <10 m). In total we use twelve Landsat path/row tiles to cover our study area (Figure 1).





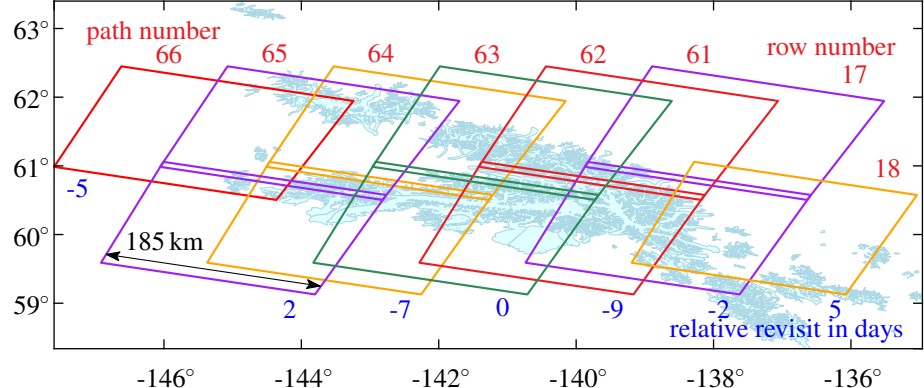

**Figure 1.** Nominal Landsat 8 footprints used over the studied region; colors are used to distinguish between different paths. Blue polygons are glacier outlines over the area of interest, stemming from the Randolph glacier inventory (Pfeffer et al., 2014).

## 2.2 Study region

The region of interest covers the Saint Elias, Wrangell and Kluane Mountain ranges, as well as some parts of the Chugach range. These ranges host roughly $42\,000\,000$ km$^2$ glacier area, whereby roughly 22% of the glacier area is connected to marine terminating fronts draining into the Gulf of Alaska. The glaciological distribution of glacier types is diverse (Clarke and Holdsworth, 2002) and there is a large precipitation gradient over the mountain range. The highest amount of precipitation falls in summer or autumn. The study area covers mountain ranges that have two different clusters of climate. Along the coast one finds a maritime climate with a small annual temperature range. These mountains function as a barrier, and the mountain ranges behind, in the interior, have therefore a more continental climate (Bieniek et al., 2012).

## 3 Methodology

GoLIVE and other velocity products are composed products from at least two acquisitions. Such velocity fields with different time spans need to be combined, in order to be of use for time-serie analysis. To reduce the noise, the temporal configuration of these products can be used to synthesize a multi-temporal velocity field.

### 3.1 Temporal network configuration

At the high latitude of the Southern Alaska scenes from adjacent tracks have an overlap of 60%. Looking at only one track (or satellite path), multiple combinations of images over time periods of integer multitudes of 16 days can be matched against each other. For example, over a 96-days period ($\Delta t$), seven images are acquired in one track and their potential pairing combinations can be illustrated as a network (Figure 2). In this network every acquisition ($I$) is a node, and these nodes are connected through



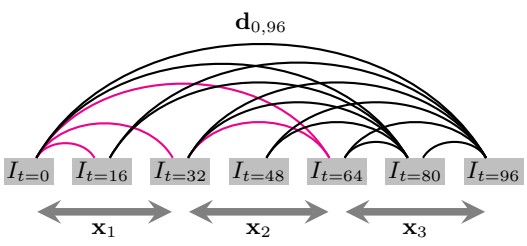

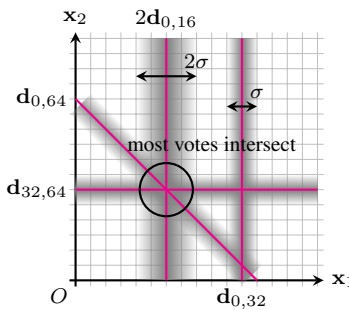

**Figure 2.** On the left is a graphical representation of Landsat 8 acquisitions ($I$) at different times ($t$) illustrated as nodes and matching solutions with displacements ($\mathbf{d}$) are shown as edges, within a network. The subscript for the displacement denotes the time interval. The velocities ($\mathbf{x}$) are estimated through the collection of all these displacements. At the right the mathematical representation or its parameter (Hough) space for voting of displacements over two time instances is shown, in this case the parameter space of only $\mathbf{x}_1$ & $\mathbf{x}_2$ are shown. The displacements ($\mathbf{d}$) translate in a solid line and the fading illustrates the different weighting to take measurement precision ($\sigma$) into account. Note the wider spread of the displacement noise for the image pair with a small interval ($\mathbf{d}_{0,16}$).

an edge that represents a matched pair leading to a collection of displacements ($d$) with an associated similarity measures ($\rho$).

When individual time steps need to be estimated, this network has in theory a great amount of redundancy. However in practice this is complicated, as combinations of images are not processed when there is too much obstruction by clouds. Furthermore, individual matches can be gross outliers due to surface change or lack of contrast and thus loss of similarity (displacement $\mathbf{d}_{0,32}$ in Figure 2). Consequently, when data from such a network is combined to synthesize one consistent velocity time series the estimation procedure needs to be able to resist multiple outliers or be able to identify whether displacement estimates could be extracted at a reliable level at all.

The network shown in figure 2 can be seen as a graph; nodes correspond to timestamps and edges to matched image pairs. Such a graph can be transformed into an adjacency matrix ($A_G$, see Figure 3). In this matrix the columns and rows represent different timestamps. The edges can be directed, meaning it can assign which acquisition is the master (reference) or the slave (search) image during the matching procedure. For the GoLIVE data, the oldest acquisition is always the reference image, hence within the matrix only the upper triangular part has filled entities. The spacing of the timesteps is 16 days and the amount of days is set into the corresponding entries when a time step is covered by an edge. Individual days are specified instead of a binary entity, to be able to merge adjacency matrices from different tracks which have different acquiring dates. If partial overlap of an edge occurs, then the time steps are proportionally distributed. For example, for a small network of three





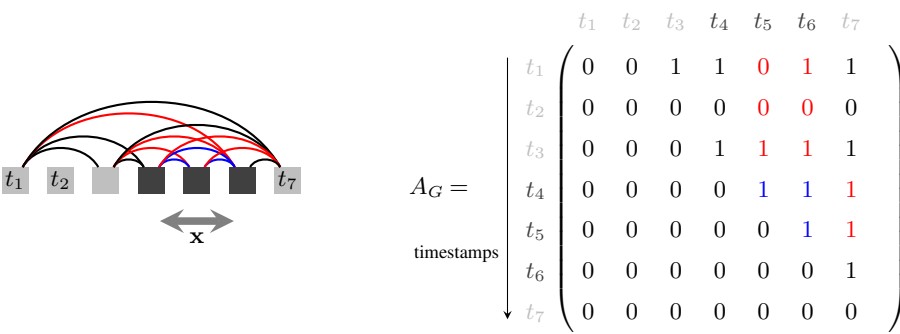

**Figure 3.** Graphical and matrix representation of a network. Here acquisition pairs within a network are illustrated and written down in an adjacency matrix ($A_G$). The dark gray squares indicate acquisitions within a period to be estimated. The connecting colors symbolize an open (red) or closed (blue) selection of displacements to be used for the velocity estimation over this period ($\mathbf{x}$).

nodes, velocity ($x$) can be estimated through least-square adjustment through the following systems of equations (Altena and Kääb, 2017a),

$$\mathbf{y} = \mathbf{A}\mathbf{x}, \text{ where } \quad \mathbf{y} = \begin{bmatrix} d_{12} \\ d_{23} \\ d_{13} \end{bmatrix}, \quad \mathbf{A} = \begin{bmatrix} \Delta t_{12} & 0 \\ 0 & \Delta t_{23} \\ \Delta t_{12} & \Delta t_{23} \end{bmatrix}, \quad \mathbf{x} = \begin{bmatrix} x_{12} \\ x_{23} \end{bmatrix}, \tag{1}$$

The construction of the temporal network makes it possible to estimate the unknown parameters, i.e. the temporal components of the velocity time series, through different formulations. This is illustrated in Figure 3, where a selection of two velocities is estimated. Displacements between the three images within this time frame can be estimated (here blue), which we here call a "closed" network. But as can be seen in the figure as red connections, other displacements from outside the time frame are over-arching and stretching further than the initial time interval. Such measurements can be of interest as they can fill in gaps, but the glacier dynamics obtained will be smeared compared to the real ones. Consequently, we call such a network configuration an "open" network (here red).

## 3.2 Voting

The velocity dataset we use (like any) contains a large amount of incorrect or noisy displacements. Moreover, a least-square adjustment is very sensitive to outliers contained in the data to be fitted. Therefore, direct estimation of velocity through the above network is not easily possible and some selection procedure is needed to exclude gross errors. Outlier detection within a network such as in equation 1 can be done through statistical testing (Baarda, 1968; Teunissen, 2000), assuming measurements ($d$) are normally distributed. However, such procedures are less effective when several gross errors are present within the set of observations. Extracting information from highly contaminated data is therefore an active field of research. For example,





robust estimators change the normal distribution to a heavy tailed distribution. Nevertheless, such estimations typically still start with normal least-squares adjustment based on the full initial set of observations, and only in the next step the weights are iteratively adjusted according to the amount of misfit. Hence, such methods are still restricted to robust a-priori knowledge or a data-set with relatively small amounts of contamination by gross errors.

Another common approach to cope with the adjustment of error-rich observations is through sampling strategies such as least-median of squares (Rousseeuw and Leroy, 2005), or random sampling and consensus (RANSAC) (Fischler and Bolles, 1981). A minimum amount of observations are picked randomly to solve the model. The estimated parameters are then used to assess how the initial model fits in respect to all observations. Then the procedure is repeated with a new set of observations.

The sampling procedure is stopped when a solution is within predefined bounds, or executed a defined amount of times after which the best set is taken. Such methods are very popular as they can handle high contamination of data (up to 50%) and still result in a correct estimate. Put differently, the break-down point is .5 (Rousseeuw and Leroy, 2005). However, we use a different approach as these methods implement polynomial models. Our data set benefits from including conditional equations as well.

Exploration into possibility theory, through the use of fuzzy-logic, is another approach towards a solution (Sun, 1994). In this study we follow this latter direction and discretize the displacement-matching search-space after which we exploit a voting strategy. In that way, a fuzzy Hough transform (Han et al., 1994) is implemented. Our matching search space is simply the linear system of equations of the network described above. To illustrate the system, an example of a network with three image

pairs (similar to equation 1) is shown in Figure 2. Every observation will fill the parameter space with a discretized weighting function. In one dimension this is a simple histogram, but in higher dimensions this will translate into a line which radially decreases in weight.

The advantage of a Hough search space is the resistance to multiple outliers. It builds support and is not reliant on the whole

group of observations. Especially, when a second or third dimensional space is used, the chances of random (line)crossing decrease significantly. Hence, such events will stand out when multiple measurements do align. Furthermore, random measurement errors can be incorporated through introducing a distribution function. In our implementation this is a Gaussian, but other functions are possible as well. The disadvantage of the fuzzy Hough transform is its limitation to implement a large and detailed search space, as the dimension and resolution depend on the available computing resources.


The fuzzy Hough transform functions as a selection process to find observations which are to a certain extent in agreement. With this selection of inliers the velocity can finally be estimated through ordinary least-squares estimation. The model is the same as used to construct the network. However, the observations without consensus (i.e. outliers) are not used. The remaining observations can, nevertheless, still be misfits, such as from shadow casting, as no ice flow behavior is prescribed in the design





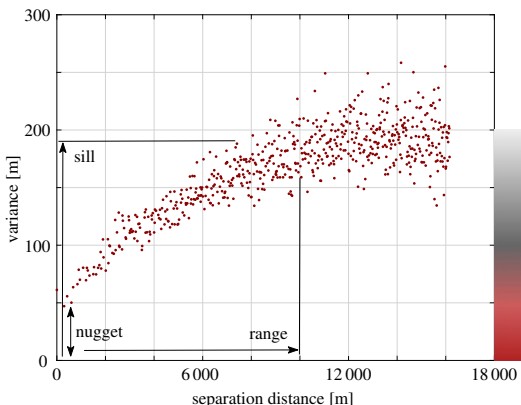

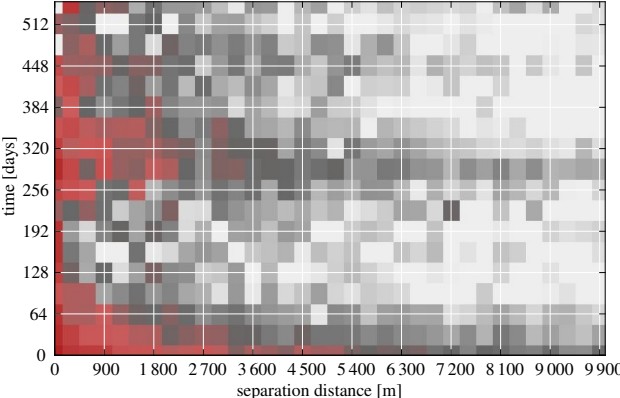

(a) Experimental variogram in the spatial domain (b) Spatio-temporal experimental variogram

**Figure 4.** Experimental variograms over a slice of the stack and over a subset of the spatio-temporal stack.The colorbar along the axis of figure 4a, is used for the coloring of figure 4b

matrix of equation 1.

### 3.3 Smoothing

Because the voting and least-squares adjustment in our implementation has no neighborhood constraints but is rather strictly

5 per matching grid point, the velocity estimates contain systematic, gross and random errors, though reduced with respect to the initial data set. Resulting in a spatio-temporal stack of velocity displacements, constructed with a regular temporal spacing that can thus be better analyzed. However, due to undersampling as a result of cloud cover, the stack might have holes. We apply a spatial-temporal smoothing taking both spatial and temporal information into account using the Whitacker approach that tries to minimize the following function,

$$10 \quad S = \sum_i w_i (\hat{x}_i - x_i)^2 + \lambda \sum_i (\Delta^2 x_i)^2. \tag{2}$$

Here $\Delta$ denotes the difference operator, thus $\Delta x_i = x_{i+1} - x_i$. Similarly, $\Delta^2$ is the double difference, describing the curvature of a signal ($\Delta^2 x_i = x_{+1} - 2x_i - x_{i-1}$). For the implementation of this method we use the procedure presented by Garcia (2010). This routine has an automatic procedure to estimate the smoothing parameter ($\lambda$) and has robust adaptive weighting ($w$). Its implementation is conducted through a discrete cosine transform (DCT), which eases the computational load. Furthermore, a

15 discrete cosine transform operates both globally and locally, and in multiple dimensions. Lastly, in order to include all data at once, the vector field is configured as a complex number field.





The smoothing parameter is operating over both the space and time dimensions, but it is a single scalar. Hence in this form it would be dependent on the choice of grid resolutions in time and space. Therefore, in order to get rid of this dependency and fulfil the isotropy property, the spatial and temporal dimensions are scaled. For this scaling estimation we construct an experimental variogram and look at its distribution (Wackernagel, 2013). Along the spatial axis, the variogram in Figure 4a shows

spatial correlation up to about 10 kilometers. This sampling interval is then used to look at the spatio-temporal dependencies, as illustrated in Figure 4b. Around a year temporal distance, one can see a clear correlation, which corresponds to the seasonal cycle of glacier velocity. From this variogram a rough scaling was estimated, and the anistropy was set towards a factor of four. In our case the pixel spacing is 300 meters and the time separation is 32 days.

**4  Results**

**4.1  Method performance**

Two different temporal networks (combinations of time intervals) can be formulated in order to calculate a velocity estimate, as is described in section 3.1. The "open" configuration includes a greater number of velocity estimates from image pairs, but this has consequences. It results in a more complete dataset, with coherent velocity fields, but when short-term glacier dynamics

occur, temporal resolution of the event may be smeared. As an example, velocity estimates from two different network configurations are shown in Figure 5 with circles highlighting selected effects. For example, a longer stretch of the speeding-up section can be seen (A) on Fisher Glacier. In the "closed" configuration, this section is slimmer and has more details. However, by including more imagery as with an open configuration, the velocity estimates are more complete (B) and are better over stable ground (C). With more displacement vectors in the configuration, smaller-scale details, such as tributaries (D), become more

apparent. Hence, depending on the application, the configuration can estimate a more complete velocity field or alternatively a temporally more precise product.

The spatio-temporal least-square estimates are still noisy or have outliers within. Therefore spatial-temporal smoothing is applied, in order to extract a better overview from the data, as is described in section 3.3. The results for one time interval are

shown in Figure 6 and again some specific details are highlighted and discussed below.

Because the surroundings of glaciers are stable or slow moving terrain is included in the smoothing, high speed-ups such as on the surge bulge on the Steele glacier (E) are dampened. They do not disappear, as the signal is strong and persistent over time, but damping does occur. Furthermore, because the smoothing makes use of the temporal domain as well, small features

emerge out of the noise such as flow tranches in the snow-covered upper parts of Kaskawulsh glacier (F). Also small areas of stable consistent fast flow like the icefall of Gates glacier, and the neighbouring staircase icefall of Root glacier (G), or the icefall between Dusty and Lowell glacier (H) get better pronounced in the smoothed data. Steps in along-track direction (I) are reduced because stable ground is motionless over time. In the along-track direction, these steps are filtered as for most of





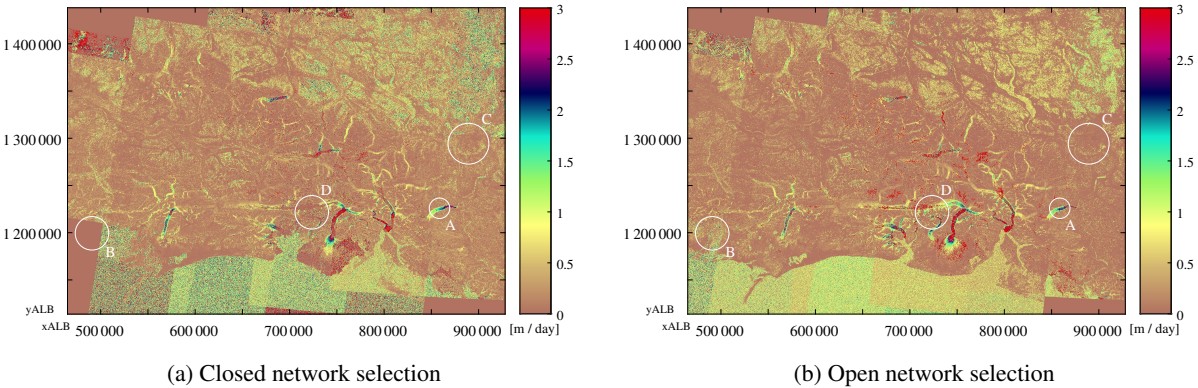

| (a) Closed network selection | (b) Open network selection |
|---|---|

**Figure 5.** Least square estimates of velocities with different network configurations, see Figure 2 for a toy example of the terminology. The study region spread over several UTM zones, hence the dataset is in Albers equal-area projection (ALB) with North American Datum 83.

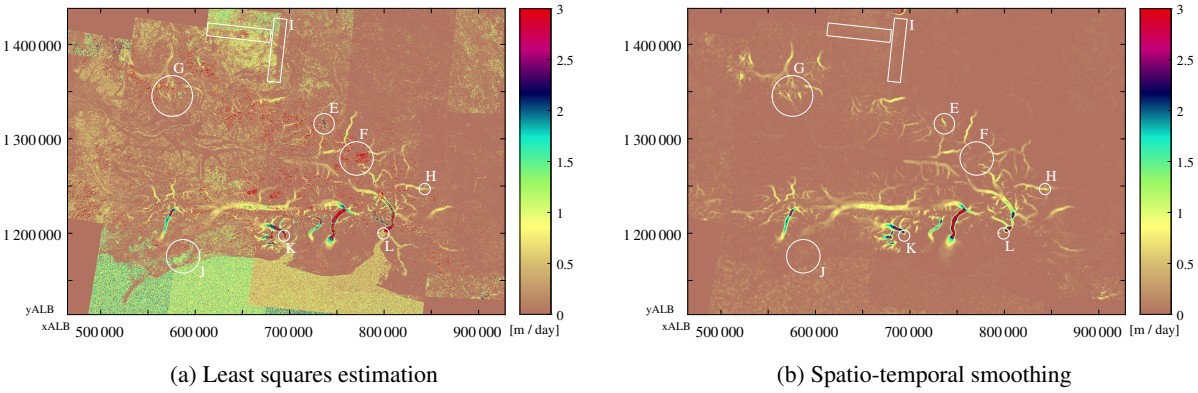

| (a) Least squares estimation | (b) Spatio-temporal smoothing |
|---|---|

**Figure 6.** Raw velocity estimates and smoothed estimates (taking space and time into account, see equation 2 and Figure 4).

the data the overlap is from the same path and thus acquired at the same time. Also random measurements over Vitus lake (J) are smoothed, as most of the estimates have a random velocity magnitude and orientation. An aspect of concern is the velocity bulge retreat of Guyot glacier (K); its front with large velocities seems to retreat in the smoothed version, while this is not the case for the original least-squares estimate. This is an effect caused by surrounding zero-valued water bodies. However, this

5  effect is not at play at every glacier, as can be seen for example at Hubbard glacier (L). In general, this small damping effect of water bodies seem to be overweighed by the advantages from a clear image.

Because the stable terrain, which has no movement, impacts the smoothed velocity estimates in particular for surge and calving fronts (i.e. for strong spatial velocity gradients), the smoothing can be supported by a glacier mask. In our case, this mask

10  is a rasterization of the Randolph Glacier Inventory (Pfeffer et al., 2014), with an additional dilation operation, to take potential advance or errors in the inventory into account. The difference in result for this masking procedure is shown in Figure 7, with





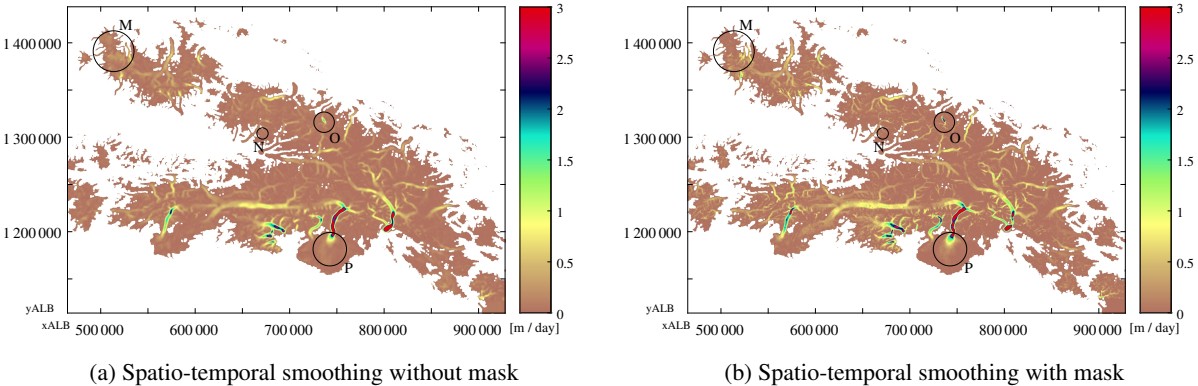

(a) Spatio-temporal smoothing without mask
(b) Spatio-temporal smoothing with mask

**Figure 7.** Smoothed grid of velocities, with masking of non-glaciated terrain (a) and with no masking (b).

some highlights. Unfortunately, speckles emerge in the estimates (M), though these only occur in the slower glacier areas. On the other hand, the masking makes it possible that individual flow branches become apparent, as can be seen for the tributaries of Barnard glacier (N). Also the surge bulges of Steele glacier (O), for example, become more pronounced. Still, the front of fast velocities of Malaspina Glacier that is reaching into the piedmont lobe (P) are taken a bit back into the Seward Threat

5  presumably because higher frequencies are given a higher weight when a mask is used. All these steps were taken to arrive at the resulting velocity sequence which reveals glaciological dynamic changes.

## 4.2 Glaciological observations

When looking at the spatio-temporal dataset some patterns that are observed by others also appear in our dataset. For example,

10  the full extent of Bering glacier slows down, as highlighted by Burgess et al. (2012), however our time series cover a period where the full deceleration towards a quiescent state can be seen. This observation of a slow down can also be made for Donjek Glacier (Abe et al., 2016) and Loogan Glacier (Abe and Furuya, 2015), see also Figure B2a for the velocity evolution at a point. In the time period covered by our study some surges appear to initiate. For example, our dataset comprises a surge traveling along the main trunk of Klutlan Glacier, see Figure B2b,B3a&B3b.

When looking at the surge occurring at Klutlan Glacier, the dataset does capture the evolution of its dynamics, as can be seen in Figure 9. The surge ignition seems to happen in the central trunk of the glacier, as the surge front progresses (with steady bulk velocities around four meters per day). The surge also propagates upwards mainly into the most westwards basin. The eastward basin does also increase in speed, but to a lesser extent, while the middle basin of this glacier system does not seem to

20  be affected significantly. Though, its extent is mostly pronounced in the lower part of the glacier, as the upward creep of velocity increase is limited and does not reach the headwalls of any basin. In Landsat imagery of late 2017, there is no indication of




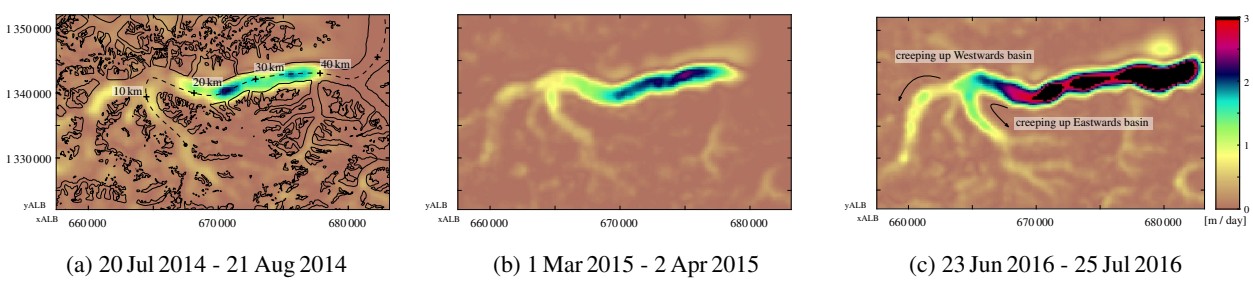

(a) 20 Jul 2014 - 21 Aug 2014          (b) 1 Mar 2015 - 2 Apr 2015          (c) 23 Jun 2016 - 25 Jul 2016

**Figure 8.** Snapshots of ice speeds at different time instances from a data compilation for the summer 2016 surge occurring on Klutlan Glacier.

any heavily crevassed terrain in the upper parts of these basins, which supports the hypothesis of a partially developed surge.

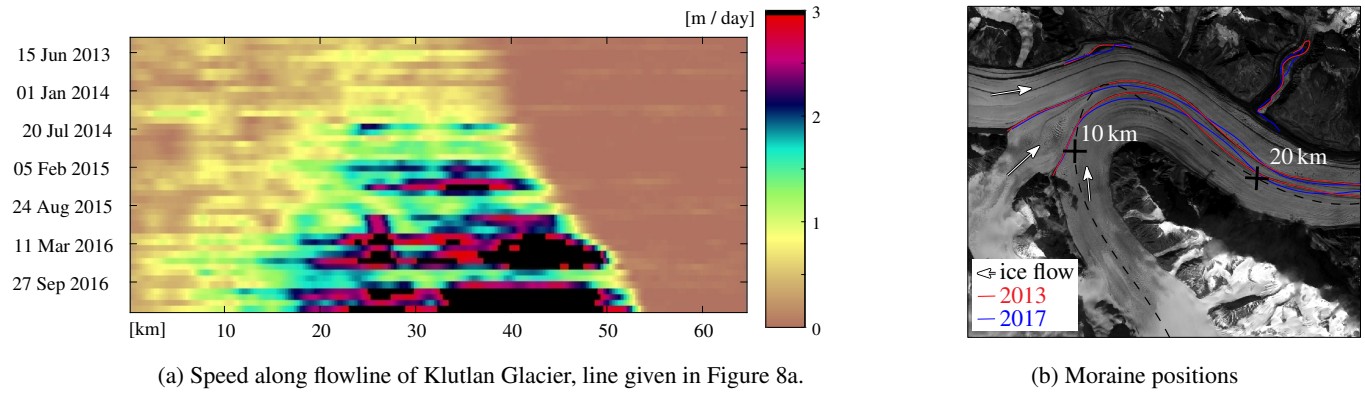

(a) Speed along flowline of Klutlan Glacier, line given in Figure 8a.          (b) Moraine positions

**Figure 9.** The speed over the central flowline of Klutlan Glacier. The markings of this flowline are shown in figure 8a. In Figure 9b the convergence of different basins of the Klutlan glacier is shown, data is from a RapidEye acquisition on the 5th of September 2013 and at the 23rd of September 2017. For comparison the 2013 image is overlain with the two morraine positions.

When looking at the velocities over the flowline of the glacier, as in Figure 9a, both the extension downstream as well as the upstream progression of the surge can be seen. Most clearly, the surge front seems to propagate downwards with a steady
5   velocity, but seems to slow down around the 50 km mark. Here, the glacier widens but the surge does continue. This suggests glacier depth is homogeneous here or glacier depth does not seem to play an important role in surge propagation.

At the end of the summer of 2016 the tributary just north of the 20 kilometer mark seems to increase in speed. This can be confirmed by tracing the extent of the looped moraines, as in Figure 9b. In the same imagery the medial moraines of the
10   meeting point of all basins are mapped as well. Here, the moraine bands before and after the event align well in the junction, indicating a steady or similar contribution over the full period. Or an insignificant effect, as the surge has not been developing





into very fast flow. In contrast, the lower part of this glacial trunk has moraine bands that do not align.

The surge behavior we observe for Klutlan Glacier is not unique and can be observed at other glaciers within the mountain ranges. For Fisher Glacier, a similar increase in speed is observed within the main trunk that later propagates downstream as

5   well as upstream. Similarly, this seems to be the case for Walsh Glacier, where a speed increase at the eastern trunk triggers a surge on the northern trunk leading to glacier-wide acceleration. On its way the fast flowing ice initiates surges in tributaries downflow, but the surge extent also creeps upslope and tributaries that were more up-glacier from the initial surge start to speed up. This is also true for Steele Glacier, that develops into a surge and Hodgson Glacier is entrained into this fast flow as well.

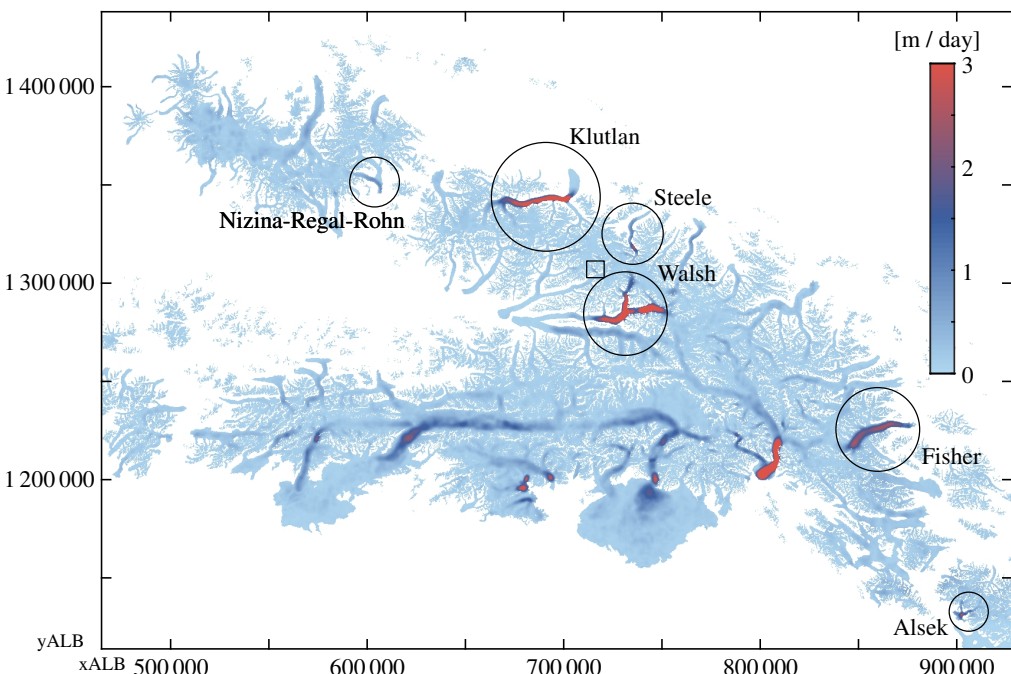

**Figure 10.** Spread of variation in flow speed over the observed period (using the difference between the 20[th] and 80[th] percentile). Different dynamic glaciers are encircled, and the square indicates the tributary glacier shown in Figure C1.

10   These analysis were best observed with the help of an animation (see supplement) but the identification was done through a simple visualization of the spread of flow speed, see Figure 10. Here the surging glaciers stand out, but also most of the tidewater glaciers, which have a highly dynamic nature at their fronts. Not only large glaciers can be identified but also the dynamics in smaller tributaries. For example, a tributary of the Chitina Glacier seems to have pushed itself into the main trunk within a two-year time-period, see figure C1.



Apart from these already known dynamics at these glaciers, additional patterns are observed. At the Nizina-Regal-Rohn glacier system in the eastern Wrangell Mountains (Figure 10), formerly negative mass balance glaciers (Das et al., 2014) appear to have generally increased in speed, but are decelerating as our data record begins. A brief period of speed-up in 2014 is followed by an extended period of low variability. These two glaciers then seem to alternate in their summer speed magnitude,

see animation and Figure B4.

## 5  Validation

### 5.1  Validation of post-processing procedures

The voting used in our procedure is assessed through validation with an independent velocity estimate. Terrestrial measure-
ments are limited in the study area, hence we use satellite imagery from RapidEye satellites over a similar timespan. Data from this constellation has a resolution of 5 meters and through processing in a pyramid fashion, a detailed flow field can be extracted. This functions as a baseline dataset to compare the GoLIVE and the synthesized data. Here we will look at a section of Klutlan Glacier, which flows from west to east, thus aligned with one of the map axis. The velocity of this glacier is, due to its surge, of significant magnitude, and therefor will have a wide spread in the voting space.

The two RapidEye images used over Klutlan Glacier were taken on 7[th] of September, and on 7[th] of October 2016. To retrieve the most complete displacement field of the glacier, we used a coarse-to-fine image matching scheme. The search window decreased stepwise (Kolaas, 2016) and the matching itself was done through Orientation Correlation (Heid and Kääb, 2012). At every step a local post-processing step (Westerweel and Scarano, 2005) was implemented, to filter outliers. The resulting
displacement field over one axis (that is x, the general direction of flow) for this period is illustrated in the top inset of Figure 11.

For the voting of the Landsat 8-based GoLIVE data, an overlapping time period was chosen, from the 11[th] of September up to the 13[th] of October 2016, nearly but not exactly overlapping with the RapidEye pair. An open configuration was used for the voting, meaning all GoLIVE displacement fields covering this time period were used, resulting in a total of 36 velocity
fields involved in the voting. The voted estimates and scores are illustrated in the third and fourth panel of Figure 11. The voted estimates have gray patches within, as these are estimates which had not enough displacement data to get a reliable estimate of equation 1, or are on stable terrain. Voting scores are high over the stable terrain, but low over the glacier trunk. To some extent this can be attributed to the surge event. The median over the stack and the median of absolute differences (MAD) are shown in the lower two panels of figure 11. These two measures are frequently used to analyze multi-temporal datasets (Dehecq et al.,
30  2015).

When looking at this time period for the GoLIVE data, a clear displacement field is shown, as both images (11th Sept., 13th Oct.) from Landsat 8 were cloud free. The pattern is in close agreement with the RapidEye version. When looking at the voted



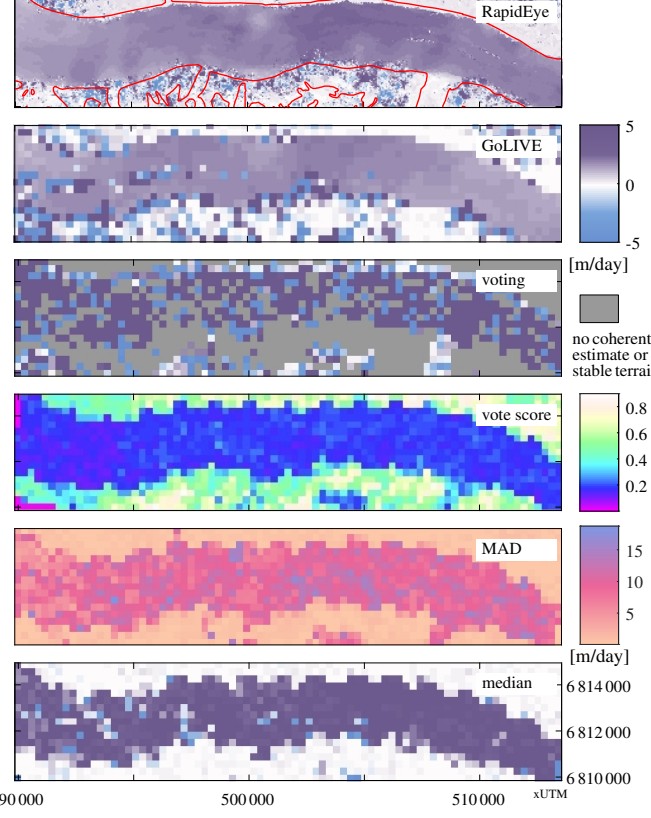

**Figure 11.** Monthly displacement in x-direction over the Klutlan Glacier using several data sources and velocity assessment schemes. The top panel shows velocities derived from two RapidEye images. Glacier borders are outlined in red. The second figure shows displacement estimates from a GoLIVE dataset (input), and the resulting voted estimate of a combination of 36 GoLIVE datasets (output). Its corresponding voting score of these estimates is shown in the fourth figure. The last two figures show the median of absolute deviation (MAD) and the median over the full dataset. These last two results would typically be used for data exploration.

estimate a similar pattern is observable but more corrupted and some data is not available. In some respect the median estimate appears to produce a better mapping. However the spread, as shown by the MAD, is considerable and will not help to justify which displacement is correct. Furthermore, the voted estimate is an estimate over a short interval, while the median estimate is calculated over the full stack.

To better assess these results, the distribution of both displacement fields are illustrated in Figure 12a and 12b. Two groups of displacement states are clearly visible, showing little movement, or a dominant movement eastwards. The voted distribution has more spread, and more outliers are present, but in general the mapping has the correct direction and magnitude. When the x-component of these displacements are compared against the RapidEye displacements a similar trend can be seen. These

10 comparisons are shown in Figure 12c and 12d, where again a similar pattern is visible. The illustrated validation do show the



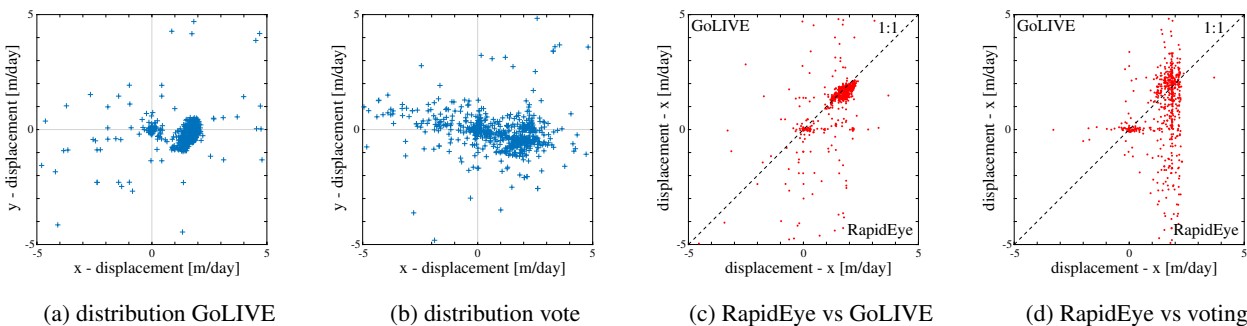

**Figure 12.** Figure 12a & 12b show the distribution of velocities for a section of Klutlan glacier, their map view are shown in figure 11. In figure 12c & 12d the same data, but now only for the x-component is set against the validation displacement.

voting scheme is able to grasp the general trend of the short term glacier flow through a large stack of corrupted velocity fields. The pair shown by for the GoLIVE dataset is one clean example, while a large extent of these velocity fields are corrupted with clouds. Hence it is a step towards efficient information extraction, though the implemented voting has many potential aspects of improvements.

## 5.2 Validation over stable terrain

A second component for validation is an analysis of the stable ground, and the effect of the smoothing of the voted estimates. The non-glaciated terrain are the locations stemming from a mask. A similar mask, also based on the Randolph glacier inventory, is used within the GoLIVE pipeline. Here, displacements over land and non-glaciated terrain are used to co-align the imagery, as geo-location errors might be present in the individual Landsat imagery. The fitting is done through a polynomial fit, still, in general these should be random, and its mean should be zero.

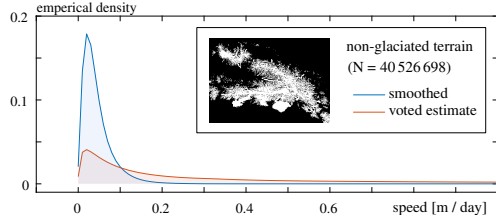

**Figure 13.** Distribution of the speed over stable terrain, for displacements extracted from the voting process, or after spatial temporal smoothing. The mask used is within the inset.

The distribution of these stable terrain measurements, more than 40 million in total, are illustrated in figure 13. Similar to the visual inspection already illustrated in Figure 6b, the distributions also show a clear improvement. This is a welcome property





as the voted estimates still seem to be noisy with significant outliers.

## 6    Discussions

Synthesized velocity estimates from our post-processing chain of GoLIVE image-pair velocity determinations are dependent
on the number and distribution of measured displacements (see Appendix A). Furthermore, the adjustment model assigns equal
weighting to individual displacements if no other information is available. Hence some velocity changes might be missed or
blurred in time. Such a drawback might be overcome with spatial constraints, such as an advection pattern imposed on the data,
although this would increase the amount of post-processing.

Another limitation of our method concerns the glacier dynamics that are constrained by our model. In the current implemen-
tation the deviation ($\sigma$) is dependent on the time interval. From a measurement perspective this makes sense, but the model
does thus not inherently account for speed change. Hence, for long time intervals the fuzzy function forces the deviation to
become slim. This reduces the ability to get a correct match, especially when glacier-dynamical changes are occurring. It might
therefore be worthwhile to explore the improvements occurring when a fixed deviation is set instead. In addition, the low score
over glaciated terrain, might indicate the deviation of the displacement is set too tight. When this deviation is given higher
bounds, the score increases and such parameter can then be used as a meaningful measure.

The smoothing parameter used is a single global parameter that assumes isotropy. In order to fulfill this property the spatio-
temporal data has been scaled accordingly. However, when severe data gaps are present, the velocity dataset still seems to
jump. This will improve when more data is available, for example by including Sentinel-2 data or incorporate across-track
matching (Altena and Kääb, 2017a). An increase in votes will result in a better population of the vote space. In addition, the
voting score, that is the consensus score in the Hough space, can be used for the initial weighting for the smoothing procedure
($w$ in equation 2). This might reduce the amount of iterations used by the robust smoother.

In the Wrangell mountain range, Armstrong et al. (2017) found a significant speed-up in summer for the lower part for most
large glaciers. We use the same data, and also see these speed-ups, but not every year. This might indicate that for these glaciers
an efficient subglacial system is not developed each year, hence storage should occur. Such storage would be in line with the
other glaciers in this region that are of surge-type. Also with our post processing it seems that signals from smaller or slimmer
glaciers such as Copper Glacier can be picked up and this glacier can be clearly identified as a speedup glacier type.





# 7 Conclusions

In the past couple of years the increase in the number of high quality optical satellite systems have made it possible to extract detailed and frequent velocity fields over glaciers, ice caps and ice sheets. The GoLIVE dataset is a repository of such velocity fields derived from Landsat 8, and available at low latency for analysis by the community. Discovery and exploration of this

resource can be complicated due to its vast and growing volume, and the complexity of spatio-temporal changes of glacier flow fields. Hence, in this study we introduce an efficient post-processing scheme to combine ice velocity data from different, but overlapping time-spans. The presented methodology is resistant to multiple outliers, as voting is used instead of testing. However, since spatial flow relations are not incorporated, the resulting synthesized time-series still have outliers. We use a datadriven spatio-temporal smoother to address this issue and enhance the visualization of real glacier flow changes.

Our synthesized time-series has a monthly (32 days) temporal interval and 300 meter spatial resolution. The time-series spans 2013 to 2017 and covers the Saint Elias Mountains and vicinity. Within this study area, we identify several surges in different glaciers at different times and their development over time can be observed. Such details can even be extracted for small tributary glaciers. Even velocities for the snow-covered upper glacier areas are in general estimated accurately.

This study is a demonstration of the capabilities of the new GoLIVE-type remote sensing products combined with an advanced data filtering and interpolation scheme. We demonstrate that our method can be implemented with ease for a large region, covering several mountain ranges. The derived smoothed time-series data contains many subtle additional changes that could be investigated. If this time-series is combined with digital elevation model (DEM) time-series (Wang and Kääb, 2015),

it becomes possible to look at changes in ice flux in great detail.

The presented velocity time-series has a high temporal dimension, especially in respect to the sensor 16-day orbit repeat cycle. Though temporal or spatial data-gaps are still present (due to the short temporal interval, cloud cover or visual coherence loss) this might partly be addressed by enlarging the temporal resolution or through additional data, such as from Sentinel-

2 (Altena and Kääb, 2017a). Fortunately, harmonization with other velocity datasets can be easily implemented, because our procedure uses only geometric information and are not dependent on sensor type. With our framework it is thus possible to make a consistent time-series composed of a patchwork of optical or SAR remote sensing products.

*Data availability.* The Global Land Ice Velocity Extraction from Landsat 8 (GoLIVE) data is available at nsidc.org/data/golive

(dx.doi.org/10.7265/N5ZP442B)



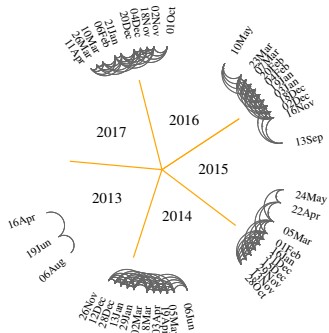

**Figure A1.** Node network of the velocity fields used in this study, for a specific path and row, 060 and 018, respectively.

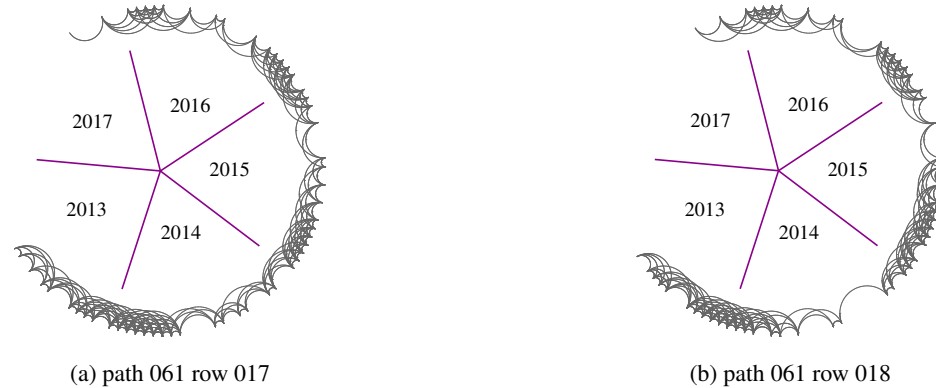

(a) path 061 row 017        (b) path 061 row 018

## Appendix A: Used velocity pairs

The GoLIVE velocity fields used in this study are of a considerable amount. In order to get an overview of the data used, the velocity pairs are plotted in data graphs. In the example of Figure A1 the graph is annotated with dates, in order to better understand the other graphs corresponding to other path and rows which are given afterwards. Within the graph each gray arch represent a displacement field used in this study. The nodes of these arches come down to certain dates, where time is oriented counter clockwise. The colored lines represent year marks and are in correspondence with the colored footprints used in Figure 1. For sake of simplicity, dates are removed from all other graphs.

## Appendix B: Corrections done by smoothing

In the following section plots are given of speed variations over selected areas of interest, the locations are denoted in Figure B1 by red crosses. Every plot has a bloxplot with the least squares estimate of a selection of observations. This selection was done




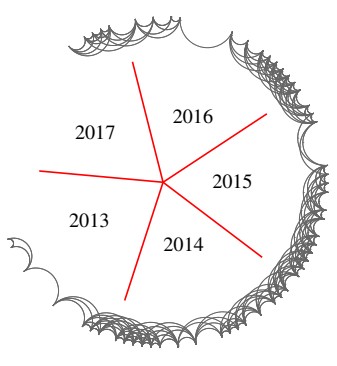
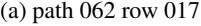

(a) path 062 row 017

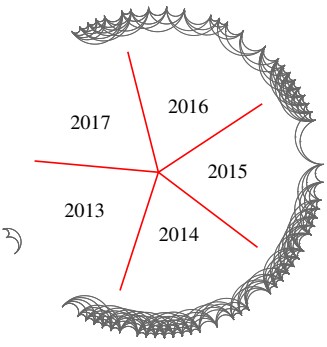

(b) path 062 row 018

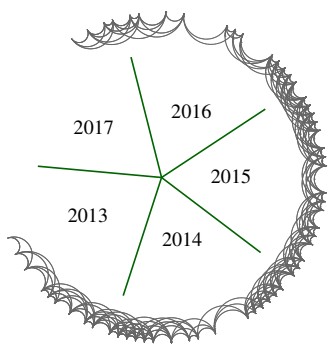

(a) path 063 row 017

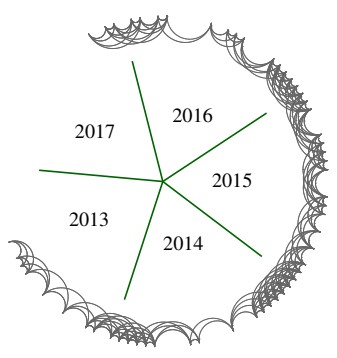

(b) path 063 row 018

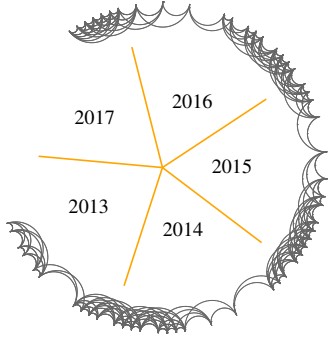

(a) path 064 row 017

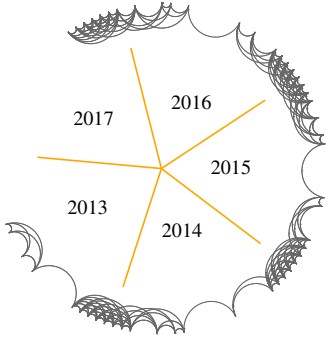

(b) path 064 row 018



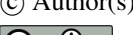

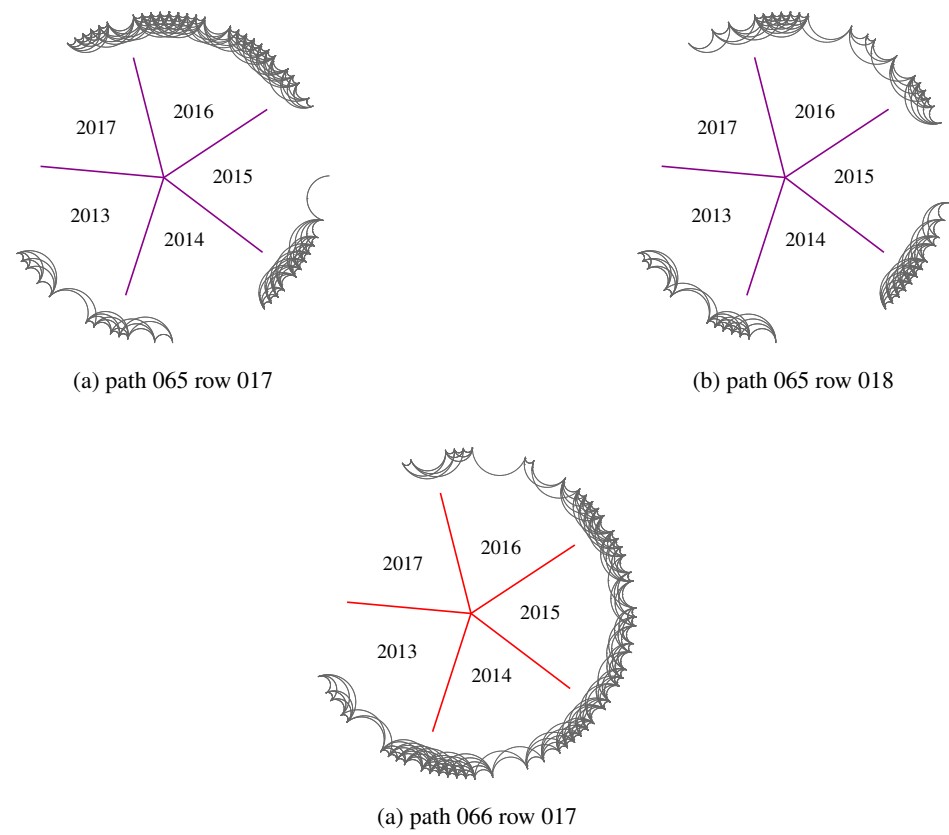

(a) path 065 row 017        (b) path 065 row 018

(a) path 066 row 017

through consensus, by voting as described in the paper. The gray lines indicate the smoothed spatio-temporal velocity. These are multiple lines, as not one estimate is taken, but a surrounding area of 5x5 pixels wide neighborhood is taken. This is done in order to have sufficient data points and see the spread of the observations and the influence of the smoother. A comparison between both estimated and smoothed version is shown in the right graph of each figure, where the white line indicates the 1:1.

## Appendix C: Tributary dynamics

From the constructed multi-temporal time-series the variance of a low and high quantile can be estimated. This gives an overview of ice masses with a highly dynamic nature. Through this simple analysis, an unknown tributary surge was identified. The push of this tributary into the medial moraine and its velocity record over time can be seen in Figure C1.





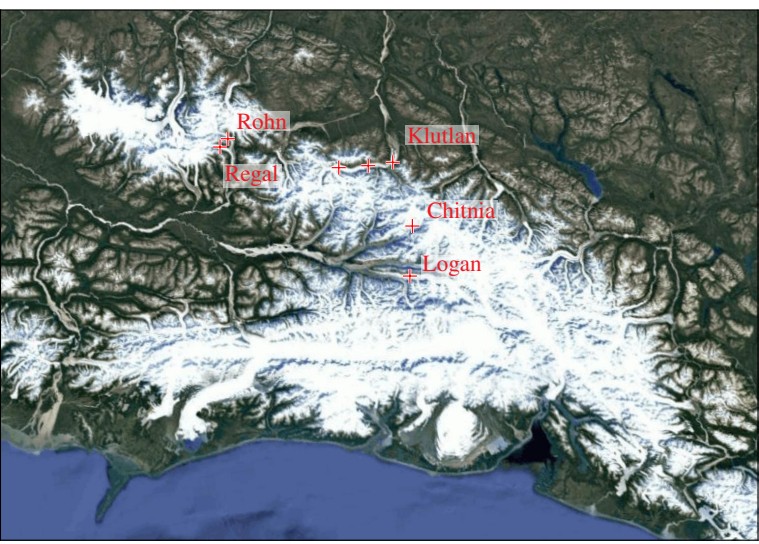

**Figure B1.** Location of the sampled points given in this section of the appendix. The background image is a Google Maps composite.

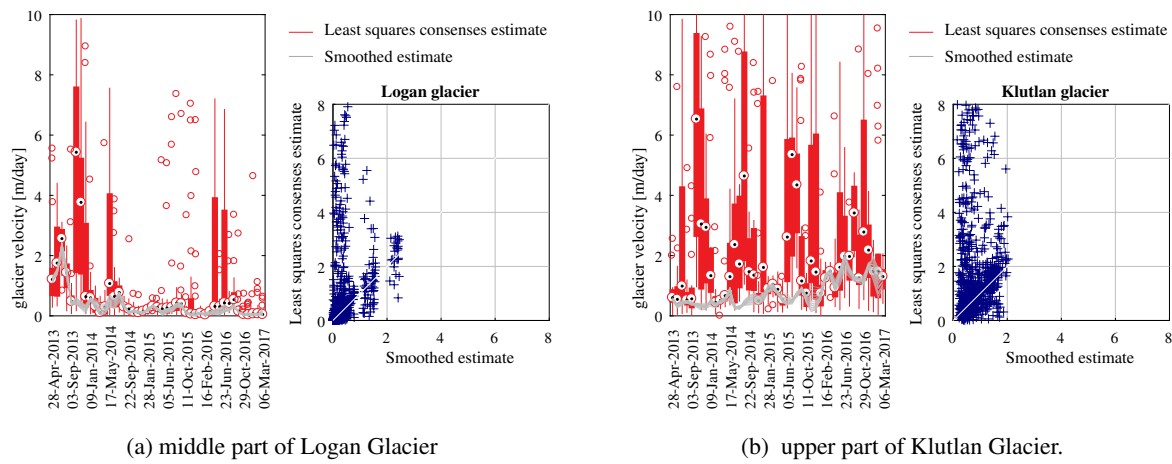

(a) middle part of Logan Glacier   (b) upper part of Klutlan Glacier.

**Figure B2.** The velocity of Logan glacier slows down and goes into quiescence, but it still seems to have some velocity increase in summer-time. For the upperpart of Klutlan the summer velocities seem to increase as the surge develops over the years.

*Author contributions.* Bas Altena led the development of this study. All authors discussed the results and commented on the manuscript at all stages.

*Competing interests.* All authors declare no competing interests.




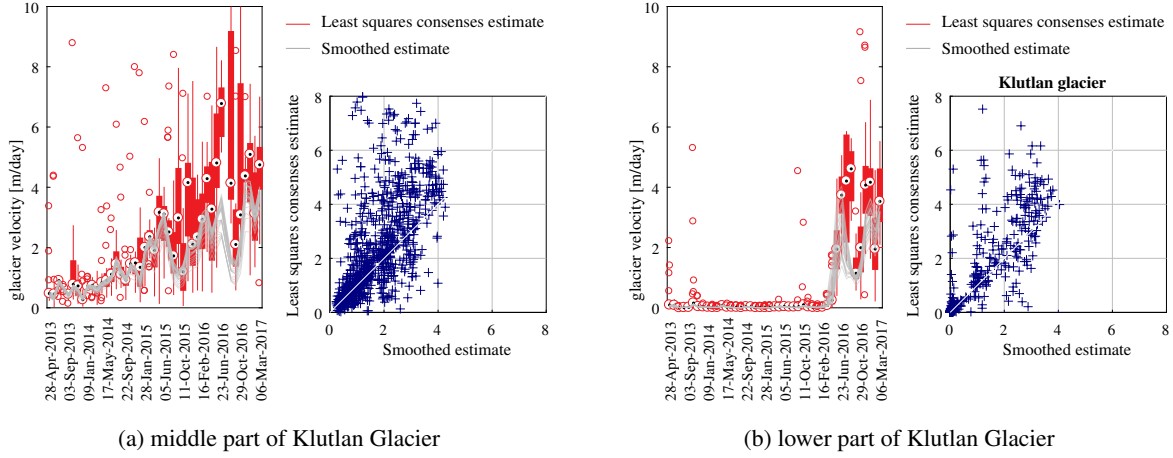

(a) middle part of Klutlan Glacier  (b) lower part of Klutlan Glacier

**Figure B3.** From our dataset the middle part of Klutlan seems to get in a surge, with heavy fluctuations, as several pulses come through. Two of these pulses seem to be visible as wel in the lower part of Klutlan glacier, which had been moving slow prior to the surge.

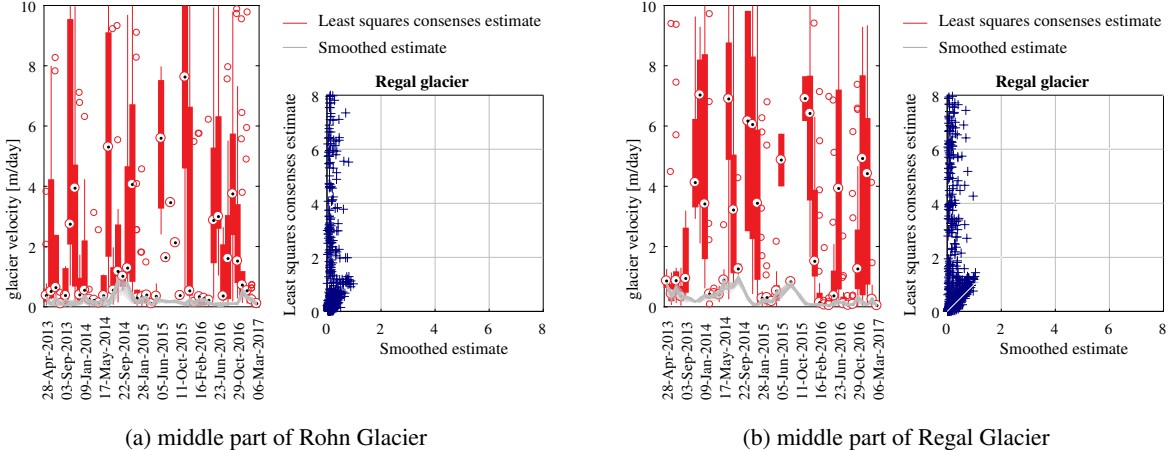

(a) middle part of Rohn Glacier  (b) middle part of Regal Glacier

**Figure B4.** Velocity evolution of the Rohn and Regal glacier that together make the Nizina glacier system. Both glaciers have every summer a small increase, however 2014 seems to stick out significantly for both. While, seems to have had another speed-up prior to 2013.

*Acknowledgements.* The research of B.A. and A.K. has been conducted through support from the European Union FP7 ERC project ICE-MASS (320816) and the ESA project Glaciers_cci (4000109873 14 I-NB). This work was supported by USGS award G12PC00066. The GoLIVE data processing and distribution system is supported by NASA Cryosphere award NNX16AJ88G. The authors are grateful to Planet Labs Inc for providing RapidEye satellite data for this study via Planet's Ambassadors Program.





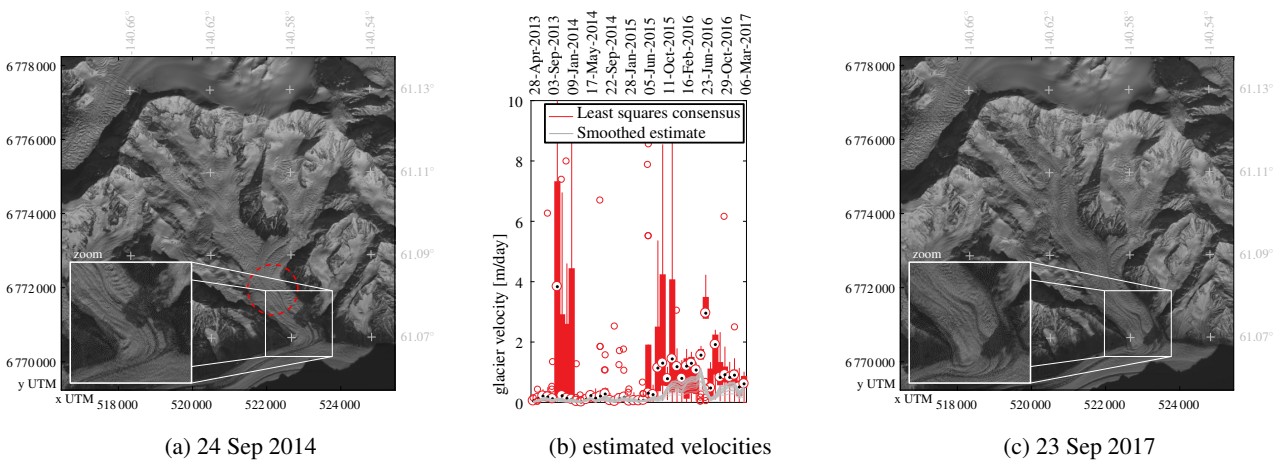

**Figure C1.** A tributary of Chitina Glacier surged in the period 2015-2016. Images are both acquired by Landsat 8, its location is indicated by a square in figure 10. The location of the time-series in figure C1b, is indicated by a red dashed circle in figure C1a.

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
