# Peer review of "Extracting recent short-term glacier velocity evolution over Southern Alaska and the Yukon from a large collection of Landsat data"

_The Cryosphere, 2018_

## Referee Comment (RC1) · Anonymous Referee #1 · 26 May 2018

General comments:

Altena et al. present a novel methodology to extract valid observations in a set of remote-sensing derived surface velocity fields, highly contaminated with noise and outliers, and to improve the visualization of short-span (32 days) velocity time series. They apply this methodology to a set of GoLive velocity fields obtained from feature-tracking of Landsat images over Alaska and period 2013-2017.

The methodology proposed in the study is novel, should help improve analysis of velocity time series, and is much needed in a context of fast-growing access to satellite-derived velocity products. The method is sound and properly referenced, but the poor

quality of the writing as well as several missing information hampers at the moment a full understanding of the methods and reproducibility. The glaciological interpretation of the results is interesting although, as mentioned by the authors, mostly discusses previously reported observations.

Specific comments:

Abtstract: You should rearrange the abstract to discuss the methodology first, and the application next. For example, l 10-13 should be moved to the end of line 4.

Section 3.1: I don't quite understand the value of AG here (except for visualization) and the connection with the following methods. What is for example the link with A in equation 1? Why is AG filled with 1 whereas the text says 'individual days are specified'? Do you assume 1-day pairs in this example?

Section 3.2: Some information is missing to fully understand the methods described here. The Hough transform proposed here is not standard and should be better introduced. At the moment, it is not possible to understand how Figure 2b is generated. For example, I understand that a pair that overlap only with X1 (resp. X2) is associated with a vertical (resp. horizontal) line, but how is the diagonal line (d32,64 - d0,32) obtained? Also introduce X1 and X2.

Section 3.3: Equation 2 needs to be fully explained, as of now, most elements are never introduced. What is x (the variable to be smoothed?), i (the pixel index?), hat operator (smoothed variable?) ? The smoothing only considers 1 dimension here, how do you consider the 3 dimensions (space + time)? I don't fully understand figure 4. Why is the variogram (4a) a scatter plot, I would expect a line. Does it show different samples or is the variability from one point to another very large? In this case, maybe use a larger step? Why are all dots of the same color, shouldn't the color scale be linked to the value of the variogram (y axis)? What is the unit of the color scale? The terms 'nugget', 'still' and 'range' are never mentioned anywhere, either introduce or delete. Rewrite the caption to clearly explain panels a and b. Finally, after reading the

variograms, I still don't understand why the 'anisotropy factor' is set to 4.

Section 4.1: A more quantitative analysis would be expected here, and some sort of conclusions. For example, lines 17-19, I would like to see more qualitative measures of how the open vs closed configuration compare in term of coverage and noise in the stable grounds. Figure 5, 6 and 7 need to be improved significantly. It is sometimes impossible to see the differences between 5a and 5b or 6a and 6b discussed in the text. The panels must be either enlarged or zoom to specific areas should be provided. Here are some suggestions: - increase the size of the panels by removing redundant axes and color scales - merge figures 5 and 6 into 1? I don't even understand the difference between panels 5b and 6a, should they not be the same? You discuss coverage, but the figures don't show any 'no data', is 0 no data? What period are the velocity fields extracted from? I would like to see a comparison with an individual velocity field here. The whole discussion is somewhat informative but remain very theoretical without any good reference field.

Section 5: This whole section should be moved before the glaciological interpretation in section 4.2.

Section 5.1: This part puzzles me. Until this section, I am convinced that the methodology is a step forward, but after this section I am not so convinced anymore. I understand the arguments that the result of the voting is obtained fully automatically and that it is limited because of gaps in the network, but (and I'm being voluntarily provocative here) if the result is not more satisfying than an individual field, why not just find a method to select the best field instead? What is the difference between figures 5, 6a and 11 and why do the results look much worse on the latter? Is a larger period considered in figures 5 and 6? Or are the results on these figures not better? In which case this confirms the need to improve figures 5 and 6, in particular enlarge and add a 'no data' color. Also the authors decided not to make this choice, how would the results change with a slight (as in not very strict threshold) filtering before the voting, or a weighting, based for example on the correlation score. I would think this help removing

a large part of outliers, in particular in cloudy or sea areas, that seem to cause a lot of gaps. On the same topic, the results for the individual field (GoLive) and median are usually filtered post-processing using for example the cross-correlations score. It would be interesting to show such results. This would probably remove outliers at the expense of coverage, giving maybe more importance to your results? This would also highly reduce the MAD. . .

Section 5.2: Again, move this section after describing the methods. P 16 last line 'the distribution also show a clear improvement'. Improvement to the voting but what about compared to the individual fields, or other methods such as median? This kind of analysis could also be used to discuss the open vs closed network.

Appendix: It took me a while to realize that the figures were described in the text. Move every description of the figures to the figures captions. Figures A should be merged together rather than being scattered on several pages. Figures B: Explain the boxplot. What is the red box (IQR?), red circle with black circle inside (median?) ? The reader does not have to guess. These figures should be improved for readability too: - simplify the boxplot, maybe show only IQR and median? - make the plots wider - for the scatter plot, change the x limits so that the scatter plot occupies the whole figure instead of $\frac{1}{4}$ of it. Since you show the 1:1 line, there is no problem in using axes with different scales.

Detailed comments:

- abstract, l 5-7: "The visualization tool. . ... as glacier surge" Keep this sentence for the main text but remove from the abstract.

- p 1, l18 "they can contribute considerably" -> "they contribute considerably"

- p3, l 9-11: this paragraph is important since it tells the reader in one line what you are going to do. Rewrite because it is too vague right now.

- section 2.1: what is the maximum time span in the GoLive products? Specify in the

text.

- p 4, l3: You mean 42 000 km2 ?

- l 4: "the glaciological distribution of glaciers is diverse" Please explain, are you talking about ocean- vs land-terminating? Size? Aspect?

- l 11: "at least two acquisitions" It is a weird statement, I assume you want to encompass the triplets in Altena and Kääb 2017? I think what is important is that the measure represents a time span (as opposed to instantaneous measure).

- l 16 "multiple combinations of . . . 16 days" Rephrase as it is a bit clumsy.

- l 19 "In this network every acquisition" -> "In this network, every acquisition". Here and in many places, a comma is missing which makes it harder to understand.

- Figure 2: Just a warning, the figure does not show correctly depending on the pdf viewer. In the caption: "At the right" -> "On the right,"

- equation 1: use v instead of x? This makes more sense since you are talking about velocity.

- p8, l 6: "Resulting in a spatio-temporal stack. . ." This sentence has no subject and does not make sense.

- equation 2: again, introduce all the terms here.

- p 9, l 26: it took me a while to understand this sentence, rephrase or cut with punctuation, in particular "surrounding of glaciers are stable or slow moving terrain" confused me.

- p 10, l 7: "stable terrain, which has no movement" a bit redundant. . .

- p 11, l17: "as the surge front progresses" It seems like the sentence was not finished. . .

- p 12, l 5: "but seems to slow down" I would add, "as shown by the break in slope" so
that the reader understand what you are referring to.

- l 6: I don't understand how you come to these conclusions, please explain or remove

- l 7: Which figure is showing the speedup?

- l 10-11 and p 13 L 1: I don't' understand these sentences. Please rephrase, some sentences are incomplete.

- figure 10, caption: mention the period here.

- figure 11: half of the color scale for the velocity is saturated, please use a different one. It also looks like the 'voting' and 'median' results are much higher than the GoLive but this might be mostly due to the choice of the color scale?

- Figure 12: panel d 'GoLive' -> 'Voting'. The title should be 'voting vs RapideEye' (y vs x) not the opposite, same for panel c.

- p 17, l 10-16: A suggestion would be to calculate a sigma that would be the (squared root) sum of the measurement error (that decreases with the time span) and the natural variability (that increases with the time span)

- p18, l 24: Here and other places, you discuss the benefits of adding data from other sensors (e.g. Sentinel). I expect velocities obtained from different sensors, with different resolution and hence sensitivity to different features, to be quite different (as shown by figure 11 between RapidEye and GoLive for example). Would that not hamper the combination of the velocities?

---

## Referee Comment (RC2) · Anonymous Referee #2 · 6 Jun 2018

This article presents algorithms to postprocess the time series of velocity data. The proposed postprocessing algorithm mainly consists of filtering and smoothing. In filtering, the authors presented voting system based on fuzzy Hough transform to select inlier. For the smoothing, Whitacker approach, accompanied by procedure by Garcia (2010) was used after making the sample space to be isotropic in both spatial and temporal perspective.

In my opinion, this methodology has unique way of approach in postprocessing the stack of spatiotemporal velocity data. The algorithms are well described, however, the article has some issues that needs to be revised. Moreover, some points claimed in

the article are not clear to me. So here I would like to provide some feedback to the authors as below:

Major concerns:

Fig. 2, right: That was the most confusing figure in this article. I think that was because I did not have enough sense about what I am looking at while I was following the article (from the beginning) and started looking at the figure. - The most confusing part is that I see 'x1' and 'x2' in horizontal/vertical axes, but I see "displacements" I suggest to revise the figure which makes the symbols more intuitive. - One thing I can suggest is to change the order of figure 2 and 3, along with the associated description. In specific, it looks like Figure 2 is more suitable to section 3.2 (voting), and Figure 3 is more suitable to section 3.1 (temporal network configuration). - In addition to this, I also suggest to move the vertical line for d_0,32. As far as I understand, d_0,32 was deviated to the right because it is outlier (as mentioned in P5, L5-6). But its position makes me confusing that the end of the line d_0,64 ends at d_0,32 in axis x1.

Figure 12: I think this figure does not sufficiently provide the information for the validation. First, I am not sure about how the Figure 12(a) and (b) would support to see how well the algorithm have worked. In specific, the distribution of the displacement or "qualitative similarity" between GoLIVE and Vote just supports the vote result "makes sense," but they cannot support how the algorithm has improved the result. Maybe the plot of displacement difference between (GoLIVE - RapidEye) and (Vote - RapidEye) might make more sense for this purpose. You can ignore this suggestion if those plots are actually the displacement difference mentioned above (but I cannot find any reason to assume they are the difference plot). Also I would suggest the authors to provide statistics about the difference. Second, I was expecting qualitative correlation between the data plotted in Figure 12(c) and (d), provided that the RapidEye velocity data is working as a reference. I think providing correlation coefficient between the data (i.e. RapidEye vs GoLIVE and RapidEye vs voting) might help supporting the validation result.

Questions / Suggestions: Figure 6: Does GoLIVE data provide any error information of the measured velocity? If so, I suggest estimating the error as well and provide that.

Figure 6(b): Based on Figure 5(a), Figure 5(b), and Figure 6(a), it looks like the up-glacier of Hubbard glacier (L) is over-smoothened, maybe because of stationary regions surrounding that. Any thoughts about this?

Figure 9(a): 1. Is the speed profile come from the raw GoLive data or filtered/smoothed result? 2. In either case, I suggest showing the time-series profile plot for both before and after the postprocessing, so that we can tell how effective the suggested algorithm was useful to investigate the glacier. 3. I don't have much knowledge about Klutlan glacier, but it looks like the 20∼50km sector of the glacier has suddenly slowed down in sometime in August-September 2016, and again suddenly accelerated after then. Is the slow-down in summer usual in glaciers in Alaska? Or is that implicates that Klutlan glacier is surge-type glacier?

Minor glitches: P2, L6: Therefor: maybe a typo? P7, L20: Figure 2 - If my understanding is correct, "three image pairs" are actually "pairs from three images"? Figure 12(d): Suggest replacing "GoLIVE" with "voting": It causes confusion with subpanel (c).

---

## Referee Comment (RC3) · Anonymous Referee #3 · 6 Jun 2018

This paper proposes a unique method in order to extract time-series glacier surface velocity data. The authors present voting system using a fuzzy voting scheme and apply the method to a set of GoLIVE glacier velocity product. They also validate their results using RapidEye-derived velocity field.

I think this automatic scheme enables us to extract full information from existing glacier velocity products. The algorithms are well written, but some points need to be revised. Here are my comments. I hope these will help to improve your manuscript.

Major comments

1: Abstract should be rewritten in order of method and application. I also think it

is better the abstract is a bit more summarized for readers to easily understand the article.

2: First paragraph in the Introduction is too simple. At least, the relation to monitor glacier velocity (content of the following paragraph) should be written.

3: Figure 2 is confusing, especially the right panel of Figure 2. I also think it is better to change the order of Figure 2 and 3.

4: You compare the results from different network, method and smoothing in Figure 5, 6, 7. You also explain the differences in some specific areas (J, K, L and so on). Please add some enlarged figures and explain what differs. I think this is important part to assess your method. Moreover, the result in Figure 7b is too smoothed. How did you consider about this?

5: The Validation section should be moved before the Glaciological observation in order to validate your result.

6: I don't understand the comparison in Figure 12a and 12b. Why did you show these Figures? Moreover, what does the cluster at 2 m/day in X axis comes from?

7: How do you access errors of velocity field in winter in terms of both snow on stable ground and GoLIVE product ?

8: The last paragraph of the Discussion is too simple. If you mention glacier dynamics deriving from your result, you should write in detail as citing some references.

Minor comments

P3L3 and elsewhere: what does "noisy" mean? Please make it clearer.

P3L7: therefore

P3L27: Randolph

P4L3-4: Please cite some references.

P11L12: Logan

P11L17: ignition → "initiation" is common to use.

P12L4-5 steady velocity → I don't think the propagation speed steady.

P12L5: the glaciers widens but the surge does continue → I don't understand what you want to tell here. Please describe in detail.

P13L6: glacier depth → ice thickness

P13L3: not unique → What does "unique" mean here?

Figure 1: Please add an explanation about "relative revisit in days"

Figure 10: What is "the observed period"?

Figure 11: Did the second figure come from raw GoLive product?

Figure 12: "GoLIVE" → "voting"

---

## Author Comment (AC1) · 14 Oct 2018

**Reviewer 1 - specific comments**

*Abtstract: You should rearrange the abstract to discuss the methodology first, and the application next. For example, l 10-13 should be moved to the end of line 4.*

Abstract has been adjusted and arranged as suggested.

*Section 3.1: I dont quite understand the value of $\mathbf{A}_G$ here (except for visualization) and the connection with the following methods. What is for example the link with $\mathbf{A}$ in equation 1? Why is $\mathbf{A}_G$ filled with 1 whereas the text says "individual days are specified"? Do you assume 1-day pairs in this example?*

The adjacency matrix is a simple matrix representation which encompasses the structure of a network. Maybe the beauty of its connectivity to equation 1 is not clearly emphasized in the manuscript. But the relational structure of displacements is similar to a leveling network. Hence, the design matrix is the incidence matrix, see section 8.3 in Strang and Borre [1997]. However, the incidence matrix is very big in our case, and it is difficult to illustrate the open (concave) and closed (convex) configuration in this way. Therefor, the adjacency matrix is used. It has now been clarified in the text.

*Section 3.2: Some information is missing to fully understand the methods described here. The Hough transform proposed here is not standard and should be better introduced. At the moment, it is not possible to understand how Figure 2b is generated. For example, I understand that a pair that overlap only with $\mathbf{x}_1$ (resp. $\mathbf{x}_2$) is associated with a vertical (resp. horizontal) line, but how is the diagonal line ($d_{32,64}$ - $d_{0,32}$) obtained? Also introduce $\mathbf{x}_1$ and $\mathbf{x}_2$.*

An additioinal paragraph about the Hough transform is added. The figure is now updated, and in addition to the toy example, also an annotation is added to clarify the relational property. This example is also highlighted in the text.

*Section 3.3: Equation 2 needs to be fully explained, as of now, most elements are never introduced. What is x (the variable to be smoothed?), i (the pixel index?), hat operator (smoothed variable?) ? The smoothing only considers 1 dimension here, how do you consider the 3 dimensions (space + time)? I dont fully understand figure 4. Why is the variogram (4a) a scatter plot, I would expect a line. Does it show different samples or is the variability from one point to another very large? In this case, maybe use a larger step? Why are all dots of the same color, shouldnt the color scale be linked to the value of the variogram (y axis)? What is the unit of the color scale? The terms "nugget", "still" and "range" are never mentioned anywhere, either introduce or delete. Rewrite the caption to clearly explain panels a and b.Finally, after reading the variograms, I still dont understand why the "anisotropy factor" is set to 4.*

The symbols are now explained in the text, and also emphasizing this is a one dimensional example. In reality the smoothing is done by taking all three dimensions into account (space + times). This is why the spatial temporal variance is plotted, this is in order to look at the amount of influence of neighboring points in space and time.

[Figure]

The figure above shows an ellipse which tries to follow the contour of equal variance. Its relation is one over four, as is tried to be visualized in this figure, this is where this anistropic correction factor comes from.

In the new manuscript the kriging terms are removed and an exponential function is plotted through the scatter.

*Section 4.1: A more quantitative analysis would be expected here, and some sort of conclusions. For example, lines 17-19, I would like to see more qualitative measures of how the open vs closed configuration compare in term of coverage and noise in the stable grounds. Figure 5, 6 and 7 need to be improved significantly. It is sometimes impossible to see the differences between 5a and 5b or 6a and 6b discussed in the text. The panels must be either enlarged or zoom to specific areas should be provided. Here are some suggestions: - increase the size of the panels by removing redundant axes and color scales - merge figures 5 and 6 into 1? I dont even understand the difference between panels 5b and 6a, should they not be the same? You discuss coverage, but the figures dont show any no data, is 0 no data? What period are the velocity fields extracted from? I would like to see a comparison with an individual velocity field here. The whole discussion is somewhat informative but remain very theoretical without any good reference field.*

We have changed the figures considerably, to make the difference more clear. Firstly, we took a timeslice out of the data-cube, which has two different data densities. That is to say, the western part has less velocity data then its eastern part. In this way the difference between both configurations can be seen more clearly. The "open" and "closed" configuration have different enhancements, and to make this more clear, zoom-ins have been created. Some of these are in the new manuscripts, others are at the end of this text. I hope these figures give a better understanding of the differences between both schemes, and their sensativity to data density.

[Figure]

[Figure]

The figures above show the histograms of velocities through least-squares consensus for the x- and y- direction. For the "closed" configuration the amount of NaN's is 17.8%, while the "open" configuration has for 8.7% no data. The distribution of the "open" configuration is more concentrated towards no-movemnt, apart from a systematic bias towards integer values. Hence, there is an improvement when the "open" configuration is used, both qualitatively and quantitatively.

*Section 5: This whole section should be moved before the glaciological interpretation in section 4.2.*
done

*Section 5.1: This part puzzles me. Until this section, I am convinced that the methodology is a step forward, but after this section I am not so convinced anymore. I understand the arguments that the result of the voting is obtained fully automatically and that it is limited because of gaps in the network, but (and Im being voluntarily provocative here) if the result is not more satisfying than an individual field, why not just find a method to select the best field instead? What is the difference between figures 5, 6a and 11 and why do the results look much worse on the latter? Is a larger period considered in figures 5 and 6? Or are the results on these figures not better? In which case this confirms the need to improve figures 5 and 6, in particular enlarge and add a no data color. Also the authors decided not to make this choice, how would the results change with a slight (as in not very strict threshold) filtering before the voting, or a weighting, based for example on the correlation score. I would think this help removing a large part of outliers, in particular in cloudy or sea areas, that seem to cause a lot of gaps. On the same topic, the results for the individual field (GoLive) and median are usually filtered post-processing using for example the cross-correlations score. It would be interesting to show such results. This would probably remove outliers at the expense of coverage, giving maybe more importance to your results? This would also highly reduce the MAD...*

Now that the bug in the code has been resolved, the new results look better than the old ones. It is not as good as a clear velocity field, but there is a consistent estimate. In this study we have explicitly moved away from methods that rely on correlation scores, as this information is already used in the image matching. Hence, we wanted to explore and exploit a different property, and approach velocity data in a fresh and new way. This is a design issue, and for glaciers it might be true the median might be very robust to get the general flow.

*Section 5.2: Again, move this section after describing the methods. P 16 last line "the distribution also show a clear improvement". Improvement to the voting but what about compared to the individual fields, or other methods such as median? This kind of analysis could also be used to discuss the open vs closed network.*
A distribution plot of the voted estimate and the best GoLIVE data out of the stack are shown in Figure 9c. It does show the voted estimate preforms worse than this *lucky pair*. Comparing this, results in a known answer, but this is not the aim of this study. Half if not more of the 2000+ velocity fields are full of errors, smearing with a median does give a meassure to compare against.

*Appendix: It took me a while to realize that the figures were described in the text. Move every description of the figures to the figures captions. Figures A should be merged together rather than being scattered on several pages. Figures B: Explain the boxplot. What is the red box (IQR?), red circle with black circle inside (median?) ? The reader does not have to guess. These figures should be improved for readability too: - simplify the boxplot, maybe show only IQR and median? - make the plots wider - for the scatter plot, change the x limits so that the scatter plot occupies the whole figure instead of 1/4 of it. Since you show the 1:1 line, there is no problem in using axes with different scales.*
Done, plots have been improved and reduced.

**Reviewer 1 - detailed comments**
*p.1 l.5-7: "'The visualization tool ... as glacier surge' Keep this sentence for the main text but remove from the abstract."*
It is removed from the abstract, and placed in the conclusions.
*p.1 l.9-11: "this paragraph is important since it tells the reader in one line what you are going to do. Rewrite because it is too vague right now."*
changed to: "Our methodology is robust as it is based upon a fuzzy voting scheme applied in a discrete parameter space, in order to filter multiple outliers".
*p.1 l.18: "'they can contribute considerably' → 'they contribute considerably'"*
done.
*sec.2.1: "What is the maximum time span in the GoLive products? Specify in the text."*
included: "At the time of writing, displacement products can cover a time interval from 16 days up to 96 days.".
*p.4 l.3: "You mean 42 000 km$^2$"*
You are right, reduced the number.
*p.4 l.4: "'the glaciological distribution of glaciers is diverse' Please explain, are you talking about ocean- vs land-terminating? Size? Aspect?"*
changed to: "The glaciers in this area are diverse, as a wide range of thermal conditions (cold and warm ice) and morphological glacier types (valley, icefields, marine terminating) occur in these mountain ranges.".
*p.4 l.5-7: "'at least two acquisitions' It is a weird statement, I assume you want to encompass the triplets in Altena and Kääb 2017? I think what is important is that the measure represents a time span (as opposed to instantaneous measure)."*
Not necessarily, therefor included another sentence to be more clear. "When imagery from multiple time instances are used, combinations of displacements, with different (overlapping) time intervals can be constructed.".
*p.4 l.16: "'multiple combinations of ... 16 days' Rephrase as it is a bit clumsy."*
changed to "the 16-day revisit makes several matching combinations of integer multitudes of 16 days possible".
*p.1 l.19: "'In this network every acquisition' → 'In this network, every acquisition'. Here and in many places, a comma is missing which makes it harder to understand."*
Adjustements have been made throughout the text.
*fig.2: "Just a warning, the figure does not show correctly depending on the pdf viewer. In the caption: 'at the right'*

→ *'On the right'."*

Caption has been changed. Also the figure was been changed, so hopefully it will now work for any viewer.

*eq.1: "use $v$ instead of $x$? This makes more sense since you are talking about velocity."*

The terminology for the unknown has been changed throughout the text, equations and figures..

*p.8 l.6: " 'Resulting in a spatio-temporal stack...' This sentence has no subject and does not make sense."*

changed to: "This least squares adjustment with voted displacements results in a spatio-temporal stack of velocity estimates that have a regular temporal spacing.".

*eq.2: "again, introduce all terms here."*

done.

*p.9 l.26: "it took me a while to understand this sentence, rephrase or cut with punctuation, in particular 'surrounding of glaciers are stable or slow moving terrain' confused me."*

changed to: "In the smoothing procedure the surroundings of glaciers, which are stable- or slow moving terrain, are included. Consequently, high speed-ups such as on the surge bulge on the Steele glacier (E) are dampened, as in this case it has a confined snout within a valley.".

*p.10 l.7: " 'stable terrain, which has no movement' a bit redundant..."*

removed "stable" and changed to "surrounding", and included "glacier velocities" for completeness.

*p.11 l.17: " 'as the surge front progresses' It seems like the sentence was not finished..."*

Added: "downwards".

*p.12 l.5: " 'but seems to slow down' I would add, 'as shown by the break in slope' so that the reader understand what you are referring to."*

Changed.

*p.12 l.6: "I don't understand how you come to these conclusions, please explain or remove"*

has been adjusted.

*p.12 l.7: "Which figure is showing the speedup?"*

The surge front is visible in figure 12.

*p.12 l.10-11: "I dont understand these sentences. Please rephrase, some sentences are incomplete."*

From a fluid mechanical point of view a moraine band is a streakline, when flow is steady streamlines and streaklines are similar. Looking at the patterns of moraines, it is therefor possible to assess, through a different concept, if the insights from the velocity fields make sense.

*p.13 l.1: "I dont understand this sentence. Please rephrase, sentence is incomplete."*

In the figure of the Klutlan surge, it is now indicated what is meant with upstream propagation.

*fig.10: "caption: mentioin the period here"*

done.

*fig. 11: "half of the color scale for the velocity is saturated, please use a different one. It also looks like the 'voting' and 'median' results are much higher than the GoLive but this might be mostly due to the choice of the color scale?"*

done.

*fig.12(d): " 'GoLive' → 'Voting'. The title should be 'voting vs RapideEye' (y vs x) not the opposite, same for panel c"*

figures have been adjusted.

*p.17 l.10-16: "A suggestion would be to calculate a sigma that would be the (squared root) sum of the measurement error (that decreases with the time span) and the natural variability (that increases with the time span)."*

True, the first term (measurement error over time) is now only included, however to put a number on the second term is challenging.

*p.18 l.24: "Here and other places, you discuss the benefits of adding data from other sensors (e.g. Sentinel). I expect velocities obtained from different sensors, with different resolution and hence sensitivity to different features, to be quite different (as shown by figure 11 between RapidEye and GoLive for example). Would that not hamper the combination of the velocities?"*

The radiometric resolution and spectral windows of Sentinel-2 and RapidEye are similar. Thus the features will have a similar *fingerprint*, however the resolution is not the same. A simple work around would be to do down-grading of the imagery.

**References**

G. Strang and K. Borre. *Linear algebra, geodesy and GPS*. Wellesley-Cambridge Press, 1997. ISBN 0961408863.

---

## Author Comment (AC2) · 14 Oct 2018

**Reviewer 2 - major concerns**

Fig. 2, right: That was the most confusing figure in this article. I think that was because I did not have enough sense about what I am looking at while I was following the article (from the beginning) and started looking at the figure. -The most confusing part is that I see "x1" and "x2" in horizontal/vertical axes, but I see "displacements" I suggest to revise the figure which makes the symbols more intuitive. - One thing I can suggest is to change the order of figure 2 and 3, along with the associated description. In specific, it looks like Figure 2 is more suitable to section 3.2 (voting), and Figure 3 is more suitable to section 3.1 (temporal network configuration). - In addition to this, I also suggest to move the vertical line for  $d_{0,32}$ . As far as I understand,  $d_{0,32}$  was deviated to the right because it is outlier (as mentioned in P5, L5-6). But its position makes me confusing that the end of the line  $d_{0,64}$  ends at  $d_{0,32}$  in axis x1. In the new version, this sub figure is changed to one figure. Its axes are now called velocity, and the measurements are called displacements. The labeling, is removed and changed into colours, and a legend is added for clarity. The figure

has moved to section 3.2 (voting). In addition to the labeling, also a toy example is included, so the positioning of the lines make sense.

Figure 12: I think this figure does not sufficiently provide the information for the validation. First, I am not sure about how the Figure 12(a) and (b) would support to see how well the algorithm have worked. In specific, the distribution of the displacement or "qualitative similarity" between GoLIVE and Vote just supports the vote result "makes sense", but they cannot support how the algorithm has improved the result. Maybe the plot of displacement difference between (GoLIVE - RapidEye) and (Vote - RapidEye) might make more sense for this purpose. You can ignore this suggestion if those plots are actually the displacement difference mentioned above (but I cannot find any reason to assume they are the difference plot). Also I would suggest the authors to provide statistics about the difference. Second, I was expecting qualitative correlation between the data plotted in Figure 12(c) and (d), provided that the RapidEye velocity data is working as a reference. I think providing correlation coefficient between the data (i.e. RapidEye vs GoLIVE and RapidEye vs voting) might help supporting the validation result.

We are aiming with this study to extract a pattern, hence the fact that the voting makes sense is the take home message. Anyways, the median of the difference is now included into the text (0.45 mtr/day for the voting and 0.27 mtr/day for the good GoLIVE pair).

**Reviewer 2 - questions & suggestions**

Figure 6: Does GoLIVE data provide any error information of the measured velocity? If so, I suggest estimating the error as well and provide that. Figure 6(b): Based on Figure 5(a), Figure 5(b), and Figure 6(a), it looks like the upglacier of Hubbard glacier (L) is over-smoothened, maybe because of stationary regions surrounding that. Any thoughts about this?

GoLIVE doesn't give error estimates for individual displacements. Some metrics are given that can be proportional or related to quality; the correlation score, peak width (precision), ratio between first and second peak height (reliability). Error propagation is possible in our framework, however then one should assume natural distributed noise, which is not the case (long tails in data).

*Figure 9(a): 1. Is the speed profile come from the raw GoLive data or filtered/smoothed result?* From the smoothed dataset, now also included in the caption of the figure.

Figure 9(a): 2. In either case, I suggest showing the time-series profile plot for both before and after the postprocessing, so that we can tell how effective the suggested algorithm was useful to investigate the glacier.

It is unclear which step the reviewer means with results before post-processing. GoLIVE data has different intervals, but if all data is used for the least-squares estimation the flowline looks like:

This is a mess, the outliers take over the estimate. It is possible, to only use the displacement with a correlation score higher than 0.6. It will result in a bit better estimates, but still very noisy:

---

## Author Comment (AC3) · 14 Oct 2018

**Reviewer 3 - major comments**

*Abstract should be rewritten in order of method and application. I also think it is better the abstract is a bit more summarized for readers to easily understand the article.*
Order has been changed.

*First paragraph in the Introduction is too simple. At least, the relation to monitor glacier velocity (content of the following paragraph should be written.*
added "Monitoring changes of ice flow is thus of importance, especially since the velocity of these glaciers seem to fluctuate considerably."

*Figure 2 is confusing, especially the right panel of Figure 2. I also think it is better to change the order of Figure 2 and 3.*
Order and figure has been changed.

*You compare the results from different network, method and smoothing in Figure 5, 6, 7. You also explain the differences in some specific areas (J, K, L and so on). Please add some enlarged figures and explain what differs. I think this is important part to assess your method. Moreover, the result in Figure 7b is too smoothed. How did you consider about this?*
Zoom-in are now included into the new manuscript.

*The Validation section should be moved before the Glaciological observation in order to validate your result.*
done.

*I dont understand the comparison in Figure 12a and 12b. Why did you show these Figures? Moreover, what does the cluster at 2 m/day in X axis comes from?*
These figures show the distribution of the velocities, if systematic effects are present in the resulting products, such can be identified. The cluster at the X-axis might be due to the inclusion of overarching velocities, where the velocities before the surge front are given a faster velocity. Or it can be a systematic error, but a traceback did not resolve in finding a direct link.

*How do you access errors of velocity field in winter in terms of both snow on stable ground and GoLIVE product ?*
I assume this is a typo ("access"→ assess). Anyways, the radiometric quality of modern day satellite imagery seem to be sensitive enough to capture features of the snowy surface so displacement estimates are possible [Jeong and Howat, 2015, Kääb et al., 2016]. An error which is present in any optical velocity field is dependent on changes in sun elevation [Berthier et al., 2005], this will be more pronounced in winter and in addition is systematic. Such effects can be filtered through simple heuristics. However, these effects are dependent on topography which has variation in different scales. Hence, it will propagate into the products, but be off less effect.

*The last paragraph of the Discussion is too simple. If you mention glacier dynamics deriving from your result, you should write in detail as citing some references.*
This is removed, as we keep the paper general, and more focussed on the processing method. The objective is to discover changes in glacier flow, and structure the data in a coherent frame, for later analysis. This is on purpose because the generated data is only surface velocities. Environmental or GIS information is not taken into account, or like ice thickness not available at this scale.

**Reviewer 3 - minor comments**

*p.3 l.3: "and elsewhere: what does 'noisy' mean? Please make it clearer."*
changed to "partly populated with erroneous velocity estimates". Noisy here means deviation with long tails, not normal-distributed or *salt & pepper* noise.

*p.3 l.7: "therefore"*
changed.
*p.3 l.27: "Randolph"*
"h" was added.
*p.4 l.3-4: "Please cite some references"*
added Molnia [2008].
*p.11 l.12: "Logan"*
removed an "o".
*p.11 l.17: "ignition → 'initiation' is common to use."*
word has been changed.
*p.12 l.4-5: "steady velocity → I don't think the propagation speed steady"*
True, the velocity pattern is a combination of patterns. This is why it starts with "most clearly". The velocity also seems to have a seasonal oscillation on top of this surge front propagation.
*p.12 l.5: "the glaciers widens but the surge does continue → I dont understand what you want to tell here. Please describe in detail."*
It seems the glacier front velocity and cross-areal distance are related (following the Bernoulli equation). This can imply that the ice thickness is constant over this part (only for some major glaciers in the region ice thickness measurements are available, see irwis data). Or alternatively, ice thickness does not matter, as it is compensated sub-glacially, or by thickness variation at the glacier surface. But ice thickness is not available for this glacier, neither have we looked at elevation time-series. However, we also found this relation at the surge front of Walsh glacier, hence it is worth it to highlight this observation. Nevertheless, more analysis are needed, to investigate this surge front dynamics.
*p.13 l.6: "glacier depth → ice thickness"*
changed.
*p.13 l.3: "not unique → What does 'unique' mean here?"*
changed to more specific formulation "not a special mechanism and similar propagation behaviour ...".
*fig.1: "Please add an explanation about 'relative revisit in days'"*
included: "The purple text colors annotates the different satellite paths of LANDSAT, while the black text indicates the relative overpass time in days in respect to path 63.".
*fig.10: "What is 'the observed period'?"*
the whole dataset, included "(2013-2018)" in the caption.
*fig.11: "Did the second figure come from raw GoLive product?"*
this figure is the best scene (manually picked) from the GoLIVE data.
*fig.12: "'GoLIVE' → 'voting'"*
changed.

**References**

E. Berthier, H. Vadon, D. Baratoux, Y. Arnaud, C. Vincent, K.L. Feigl, F. Remy, and B. Legresy. Surface motion of mountain glaciers derived from satellite optical imagery. *Remote Sensing of Environment*, 95(1):14–28, 2005.

S. Jeong and I.M. Howat. Performance of Landsat 8 Operational Land Imager for mapping ice sheet velocity. *Remote Sensing of Environment*, 170:90–101, 2015.

A. Kääb, S.H. Winsvold, B. Altena, C. Nuth, T. Nagler, and J. Wuite. Glacier remote sensing using Sentinel-2. Part I: Radiometric and geometric performance, and application to ice velocity. *Remote sensing*, 8(7):2072–4292, 2016. doi: 10.3390/rs8070598.

B.F. Molnia. *Glaciers of North America-Glaciers of Alaska*. Number 1386-K. Geological Survey (US), 2008.

---

## Author Response (AR1)

**Response to review**

Sapporo, Hokkaido
bas.altena@geo.uio.no

October 14, 2018

Dear editor and reviewers,

Thank you very much for your thorough assessment of the manuscript. I have been going through your comments, and in doing so found some lines of code which where misplaced. The orientation in the Hough transform for combined velocities was incorrect[1]. Hence, the results in the manuscript highlight only the displacements which had whole numbers. Thus the theory is correct, but the implementation was not optimal. Basically, the manuscript was a slimmed version of the correct implementation; "an arrow can lose its feathers but not its point"! Even with limited amount of data, information can be extracted.

Nevertheless, this typo resulted in redoing the study, which took some time, in addition to the holiday season. Hence, I hope you understand this delay and enjoy the new results which have higher detail and more information within. In the pages hereafter inset are given, which are of interest to all referees, followed by the corrections made.

Sincerely,

Bas Altena
also on behalf of Ted Scambos,
Andreas Kääb & Mark Fahnestock
* * *
[1]The diagonal lines were not placed correctly in the Hough space. Consequently, only horizontal and vertical lines were able to identify an intersection.

[Figure]

figure 1: Insets of different sections, that a shown hereafter.

[Figure]

figure 2: Seward Glacier.

[Figure]

(a) Smoothing without a mask

[Figure]

(b) Smoothing with a glacier mask

[Figure]

figure 4: Fischer Glacier.

[Figure]

(a) Closed configuration        (b) Open configuration

(a) Smoothing without a mask        (b) Smoothing with a glacier mask

[Figure]

figure 7: Steele Glacier.

[Figure]

(a) Closed configuration  (b) Open configuration

(a) Smoothing without a mask  (b) Smoothing with a glacier mask

[Figure]

figure 10: Kaskawulsh Glacier.

[Figure]

(a) Closed configuration

(b) Open configuration

(a) Smoothing without a mask

(b) Smoothing with a glacier mask

[Figure]

figure 13: Kennicott Glacier.

[Figure]

(a) Closed configuration

(b) Open configuration

(a) Smoothing without a mask

(b) Smoothing with a glacier mask

[Figure]

figure 16: Lowell Glacier.

[Figure]

(a) Closed configuration

(b) Open configuration

(a) Smoothing without a mask

[Figure]

(b) Smoothing with a glacier mask

[Figure]

figure 19: Guyot Glacier.

[Figure]

(a) Closed configuration

(b) Open configuration

(a) Smoothing without a mask

[Figure]

(b) Smoothing with a glacier mask

[Figure]

figure 22: Hubbard Glacier.

[Figure]

(a)  Closed configuration

(b)  Open configuration

(a)  Smoothing without a mask

(b)  Smoothing with a glacier mask

[revised manuscript text omitted]

$$
A_G = \quad
\begin{array}{c}
\\
t_1 \\ t_2 \\ t_3 \\ t_4 \\ t_5 \\ t_6 \\ t_7
\end{array}
\begin{array}{ccccccc}
t_1 & t_2 & t_3 & t_4 & t_5 & t_6 & t_7 \\
\left(\begin{array}{ccccccc}
0 & 0 & 1 & 1 & 0 & 1 & 1 \\
0 & 0 & 0 & 0 & 0 & 0 & 0 \\
0 & 0 & 0 & 1 & 1 & 1 & 1 \\
0 & 0 & 0 & 0 & 1 & 1 & 1 \\
0 & 0 & 0 & 0 & 0 & 1 & 1 \\
0 & 0 & 0 & 0 & 0 & 0 & 1 \\
0 & 0 & 0 & 0 & 0 & 0 & 0
\end{array}\right)
\end{array}
$$

~~(yshift=3.2exharrowleft) – (yshift=3.2exharrowright) nodemidway,aboveimage timestamps; (yshift=1.5ex,xshift=-1.7exvarrowtop) – (xshift=-1.7exvarrowbottom) nodenear end,lefttimestamps; 
[revised manuscript text omitted]

---

## Author Response (AR2)

**Response to editor review on 30th of October**
* * *
Oslo, Norway
bas.altena@geo.uio.no

November 1, 2018

Dear editor,

Thank you very much for your thorough check of the manuscript. The highlighted adjustments you pointed out have been adopted in the new manuscript, which is provided with this new submission. Hopefully, the manuscript can now be send out for a second review round.

Sincerely,

Bas Altena
also on behalf of Ted Scambos,
Andreas Kääb & Mark Fahnestock

---

## Author Response (AR3)

**Response to review with minor comments**

Castricum, Netherlands
bas.altena@geo.uio.no

January 8, 2019

Dear editor and reviewers,

Thank you very much for your effort to assess our manuscript. Our apologies for the tedious work you as editor and reviewers had to do, concerning the spelling errors. We have adjusted the minor comments as given by you, and other small errors and spelling has been adjusted, please see the marked-up manuscript.

However not all comments are implemented, specifically one issue, in the following we will give our reasons why we did not. Reviewer #4 points out to an issue which is common for inter-seasonal velocity extraction with optical imagery; shadows in winter cause the pattern matching to fix on the shadows, as an example see figure 1. We have been thinking about how to quantify this issue, but the study in its current form, makes it difficult to implement. There are several reasons why we have not included this aspect in this manuscript:

[Figure]

(a) Landsat 8 acquisition on the 8$^{th}$ of August 2016      (b) GoLIVE displacement in March 2014 [m/day].

figure 1: An example of surface displacement of glacier ice (soft colours) and shadow displacements (red) over Walsh Glacier.

The major issue we see is the sparse connection of velocity products. The network in the appendix already shows this issue, where large gaps exist throughout the season. Here one can see that in general the winter velocities are better constrained, while the summer has less coverage. This might result in ill-conditioned estimation of velocity over the summer season. It is a limitation of the GoLIVE product, which has only velocity products ranging up to 96 days. It is not only shadow tracking in winter, which can be a problem. Snow cover and melt-out (which occur in autumn and spring) are therefore the most sensitive time periods for similarity loss. Thus the presented results in this study are on the edge of signal-to-noise, as the redundancy-number is generally low.

Secondly, in the current form the misfit or singularity of the least squares adjustment is not saved. This would be a good indicator for precision, however this would mean the calculation would have to be re-run. In addition it will be questionable if disentangling these factors (correlation, coverage, coherence) is possible. We are working on a follow-up study that does included multiple sensors (Sentinel-1&2) to get longer time-spans and multi-modal velocities. In this way, thorough decomposition and analysis is possible, but this would require re-framing the whole manuscript, and switching the objective of the study. Shifting emphasis from discovery to analysis-ready-data.

Thirdly, the shadow error is dependent on the product used. Normalize cross-correlation is most sensitive to large intensity variation [Debella-Gilo and Kääb, 2011], which the GoLIVE workflow is based upon [Fahnestock et al., 2016]. To reduce this effect the GoLIVE correlator uses imagery which have gone through a high-pass filter [Scambos et al., 1992]. Setting the wavelength of the filter is a subtle trade-off, as at the same time shading and shadowing of smaller surface topography is a feature to correlate (especially in winter). In modern day satellite instruments information is present in the imagery over shadow casted terrain, hence in principle frequency based orientation correlation [Heid and Kääb, 2012] might perform better for this specific issue. To wrap-up, to us this issue more dependents on the input and less on the presented methodology.

The verification as given through the rapidEye pair over Klutlan Glacier could be a solution to assess the influence of shadowing. A different region should be choosen in the upper part of a glacier. But this would mean multi-spectral matching, different processing steps and thus additional information about such steps need to be included into the text. It would deviate from the subject of the paper, but practically it is possible.

The given signal of the shadows in the GoLIVE products might be filtered out through locatization through ray-casting of the sun with the help of a digital elevation model. However, again this is not the scope of the study and as is, the product looks at displacements through surface appearance. Thus the created framework will follow features of the glacier surface and shadow cast, as both comply with the presented framework and do not violate the geometric property. Neighborhood operators (like the common median filter [Paul et al., 2015]) also fail to grasp this effect, and to our knowledge this is still an open issue. Hence, any previous published work will have this signal within, not only displacement fields, but also elevation models derived from optical sensors [Berthier et al., 2005].

To be clear, it is by no means our intention to downplay the concern of the reviewer, as the issue raised by the reviewer is of much interest. But solving or nailing down this issue will result in considerable work, while we think the resulting analysis in the context of this work will not be strong, they might give a hint, but not much insight. If needed, we will do this, but it would mean more time to do so. On the other hand, we hope with the given arguments above you understand the decision not to do so. The claims in our manuscript are intentionally not bold and we hope it shows this direction is work-in-progress, and much can be improved.

Sincerely,

Bas Altena
also on behalf of Ted Scambos, Mark
Fahnestock & Andreas Kääb

**Reviewer #4**

In the following some specific responses are given to comments given by the reviewers. Missing remarks have been adjusted as suggested, as can also be seen in the new manuscript.

*P1, L16: The Gardner et al. (2013) paper is a bit outdated now, as it only considers data up to 2009. A paper such as Harig and Simons (2016) is better as it brings the s.l. contribution record up to 2014.*
Thank you, the study was unknown to me.

*P4, L6-L8: it would be useful to provide a reference, and perhaps some specific numbers, to back up your statements here. This paper could help: Fleming et al. 2000*
Again, thanks for pointing this out, the study is now included.

*Fig. 3: I don't understand why the two d1,2 displacements (purple and blue arrows) are different from each other. Please explain.*
A more descriptive piece is added.

*P8, L7: the lines are described as being blue, purple and yellow, but they look to be blue, purple and orange to me*
You are right, the color is "gold", maybe yellow is most suited than.

*P9, L5: you mention undersampling as a result of cloud cover, but presumably undersampling could also occur due to snow cover (particularly from new snowfall between image acquisitions)? This follows on from my major comment above.*
True, this mention is now included in the text, again this does not the affect the performance of the methodology, as argumented above.

*Fig. 5: unclear what needs to extent means in the figure caption. Perhaps this is meant to be written as "needs to extend"?*
Changed to enlarged, which might be better to describe the scaling.

*P11, L19: it would be useful to provide the distance of the velocity bulge retreat to help the reader understand how significant it is*
We have included a flowline time series of the outlet of Hubbard Glacier. It shows our choosen timestep is at the upper limit of resolution, the difference between signal and noise becomes very difficult. More or different data will clearly improve confidence in the product.

*Fig. 12a: can you make a comment in the paper as to whether the horizontal stripes in the velocity pattern are real? E.g., are these due to seasonal variability in motion during the surge? Or lack of good velocity matches during the winter?*
See former comment.

*P20, L18: missing figure number (??). Also reword this sentence, as I have a hard time trying to figure out its meaning: "In the example of Figure ?? the graph is annotated with dates, in order to better understand the other graphs corresponding to other path and rows which are given afterwards."*
Sorry, this piece of text was not updated yet, and described the former graphs. It is now updated.

[revised manuscript text omitted]

---

## Author Response (AR4)

**Response to editor review on 31[st] of January**

Oslo, Norway
bas.altena@geo.uio.no

January 31, 2019

Dear editor,

Thank you very much for highlighting the minor remarks of the manuscript. These have been adjusted accordingly. In addition, we have included a paragraph about the shortcomings of velocity fields from optical datasets. The content of the text is to a large extent in the same form as the response we gave to the reviewer. This text is situated in the manuscript as the last paragraph of the discussion, just above the conclusion section. With these adjustments, we hope the work is now acceptable for publication.

Sincerely,

Bas Altena
also on behalf of Ted Scambos,
Andreas Kääb & Mark Fahnestock

---

## Author Response (AR5)

Sem Sælandsvei 1, Oslo, Norway
bas.altena@geo.uio.no

February 18, 2019

Dear editor,

Thank you very much for accepting our manuscript. In the following a track-changes of the latest manuscript is included, where the latest comments have been implemented.

Sincerely,

bas altena

[revised manuscript text omitted]